# Doubly robust identification of treatment effects from multiple environments

**Piersilvio De Bartolomeis**[1], **Julia Kostin**[1], **Javier Abad**[1], **Yixin Wang**[2], **Fanny Yang**[1]
[1]ETH Zurich  [2]University of Michigan

## Abstract

Practical and ethical constraints often require the use of observational data for causal inference, particularly in medicine and social sciences. Yet, observational datasets are prone to confounding, potentially compromising the validity of causal conclusions. While it is possible to correct for biases if the underlying causal graph is known, this is rarely a feasible ask in practical scenarios. A common strategy is to adjust for all available covariates, yet this approach can yield biased treatment effect estimates, especially when post-treatment or unobserved variables are present. We propose RAMEN, an algorithm that produces unbiased treatment effect estimates by leveraging the heterogeneity of multiple data sources without the need to know or learn the underlying causal graph. Notably, RAMEN achieves *doubly robust identification*: it can identify the treatment effect whenever the causal parents of the treatment or those of the outcome are observed, and the node whose parents are observed satisfies an invariance assumption. Empirical evaluations across synthetic, semi-synthetic, and real-world datasets show that our approach significantly outperforms existing methods[1].

## 1 Introduction

Treatment effects are key quantities of interest in applied domains such as medicine and social sciences, as they determine the impact of interventions like novel treatments or policies on outcomes of interest. To achieve this goal, researchers often rely on randomized trials since randomizing the treatment assignment guarantees unbiased treatment effect estimates under mild assumptions. However, methods relying on randomized data face several issues, such as small sample sizes, sample populations that do not reflect those seen in the real world, and ethical or financial constraints. As a result, there is growing interest in using observational data to estimate treatment effects.

A fundamental challenge in using observational data is the selection of a *valid adjustment set*, i.e. a set of covariates that can be used to identify and estimate the treatment effect. Although criteria for identifying valid adjustment sets are well-established, they rely on the knowledge of the underlying causal graph. When the graph is not known, practitioners often adjust for all available covariates (Austin, 2011). Yet, this approach runs the risk of including *bad controls*—covariates that open backdoor paths between the treatment ($T$) and the outcome ($Y$), thereby introducing bias into the treatment effect estimate. For instance, consider the causal graphs illustrated in Figure 1, where $\{X_1, X_2\}$ are the observed covariates. In Figure 1a, both $X_1$ and $X_2$ are parents of $T$, and adjusting for $\{X_1, X_2\}$ blocks all backdoor paths between $T$ and $Y$, making it a valid adjustment set. In contrast, in Figure 1b, $X_1$ is a child of $T$, and conditioning on it opens a backdoor path between $T$ and $Y$, introducing bias in the effect estimate. In the latter case, $X_1$ is referred to as a *bad control*.

Bad controls pose a significant challenge when the causal ordering of the observed covariates is not clear (King, 2010; Montgomery et al., 2018). A prominent example for a bad control is the birth-weight paradox (Wilcox, 2001) from which it was concluded that when estimating the effect of maternal smoking ($T$) on infant mortality ($Y$), the birth weight would be a bad control like $X_1$ in Figure 1b, as it is a child node of $T$ and likely leads to collider bias. Further, Acharya et al. (2016)

---

[1]See our GitHub repository: `https://github.com/jaabmar/RAMEN/`

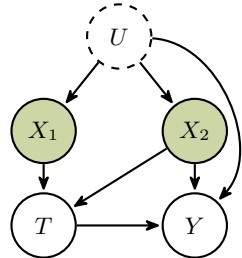
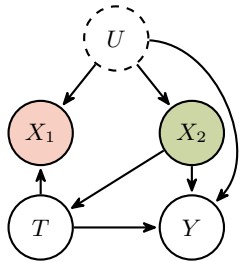

(a) $\{X_1, X_2\}$ is a valid adjustment set

(b) $\{X_1, X_2\}$ is not a valid adjustment set

Figure 1: Two causal graphs illustrating when the set of all covariates is or is not a valid adjustment set: (a) $\{X_1, X_2\}$ blocks all backdoor paths between $T$ and $Y$, making it a valid adjustment set; (b) $X_1$ opens a backdoor path between $T$ and $Y$, introducing bias in the treatment effect estimate if adjusted for. Unobserved variables are dashed and colored in white, bad controls are colored in red, and *good controls*—covariates that can be included in the adjustment set—are colored in green.

found that up to two-thirds of empirical studies in political science that make causal claims inadvertently include bad controls in their analysis, leading to biased treatment effect estimates. Several works have tried to tackle the problem of bad controls using expert-driven structural knowledge, e.g. by leveraging anchor variables (Cheng et al., 2022a; Shah et al., 2022), but such domain expertise is often unavailable in practice. Shi et al. (2021) propose an alternative approach that leverages access to multiple heterogeneous data sources—e.g. observational studies from different countries—to identify and estimate the treatment effect in the presence of bad controls. However, their approach can fail to identify the treatment effect when some variables in the causal graph are unobserved and their distribution shifts across environments, e.g. for the causal graph illustrated in Figure 1b.

In this work, we propose **R**obust **ATE** identification from **M**ultiple **EN**vironments (RAMEN ♻), an algorithm that identifies and estimates the average treatment effect in the presence of bad controls without the need to know or learn the complete causal graph. Notably, RAMEN leverages the heterogeneity of multiple data sources to achieve *doubly robust identification*: it can identify the average treatment effect whenever the causal parents of the treatment or those of the outcome are fully observed, and the node whose parents are observed satisfies an invariance assumption. In particular, our methodology relaxes the full observability requirements of Shi et al. (2021), requiring only partial observability of the causal graph. Our key contributions are outlined below.

- We propose the first algorithm, to our knowledge, that leverages multiple heterogeneous data sources to identify and estimate the average treatment effect in the presence of both bad controls and unobserved variables. Our algorithm is based on a novel double robustness property that offers two strategies to identify the treatment effect: either observe the causal parents of the treatment or those of the outcome.

- We demonstrate that our algorithms significantly outperform existing approaches for treatment effect estimation in the presence of bad controls on synthetic, semi-synthetic and real-world datasets. We further evaluate our method on a real-world example, showing that our results align with established epidemiological knowledge.

## 2 RELATED WORK

Various criteria and methods have been proposed for covariate selection, often in the form of necessary and sufficient conditions for a given causal graph, such as the backdoor criterion and its variations (Pearl, 1995; Shpitser et al., 2010; Vander Weele & Shpitser, 2011; Maathuis & Colombo, 2015; Perković et al., 2018). However, since the causal graph is rarely known in real-world applications, the most common heuristic approach assumes that all observed covariates are pre-treatment and includes all of them (Austin, 2011, p. 414). Yet, including all covariates has several drawbacks: certain pre-treatment covariates can introduce M-bias (Entner et al., 2013; Gultchin et al., 2020; Cheng et al., 2022b; Shah et al., 2022), and even when bias is not an issue, selecting a smaller subset

of covariates leads to more efficient estimates (Hahn, 2004; White & Lu, 2011; De Luna et al., 2011; Rotnitzky & Smucler, 2020; Witte et al., 2020; Henckel et al., 2022; Guo et al., 2023).

The problems described above are orthogonal to our focus in this paper, which is on scenarios where bad controls are present and the underlying causal graph is unknown (see Appendix B for a complete literature review). In this context, previous works have achieved partial identification, albeit with significant computational costs (Hyttinen et al., 2015; Malinsky & Spirtes, 2017). More recently, several works have proposed methods for point identificatios using expert-driven structural knowledge. Cheng et al. (2022a) rely on a known anchor variable, Shah et al. (2022) assume that a direct parent of the treatment variable is known, and Shah et al. (2024) assume all children of the treatment variable are observed and known. However, a significant limitation of these approaches is their dependence on structural knowledge of the causal graph, which is often unavailable in practice. In contrast, our methodology achieves point identification by leveraging multiple heterogeneous data sources, effectively circumventing the need for partial knowledge of the underlying causal graph.

Finally, our notion of double robustness differs significantly from classical results in estimation (Robins et al., 1994; Vansteelandt et al., 2008; Chernozhukov et al., 2018) and identification (Arkhangelsky & Imbens, 2022), which focus on robustness to model misspecification. A more similar concept is robust identification in instrumental variable settings (Kang et al., 2016; Hartwig et al., 2017; Guo et al., 2018; Kuang et al., 2020; Hartford et al., 2021), which allows for a fraction of instruments to be invalid. In contrast, our method guarantees identification that is robust to unobserved variables when either (i) the observed covariates include all parents of $T$ and $T$ satisfies invariance assumptions, or (ii) the same holds for $Y$ – hence yielding *double robustness*.

## 3  PROBLEM SETTING

For a fixed directed acyclic graph (DAG) $\mathcal{G}$, we denote the complete set of its nodes by $\tilde{Z}$ and the *observed* nodes by $Z$. We denote the index set of parents, ancestors, and descendants for any node $\tilde{Z}_i$ by $\mathrm{Pa}(\tilde{Z}_i)$, $\mathrm{An}(\tilde{Z}_i)$, and $\mathrm{De}(\tilde{Z}_i)$, respectively. Additionally, for an index set $S$, $\tilde{Z}_S$ denotes the subvector of $\tilde{Z}$ corresponding to the indices in $S$. We assume the data is collected under different conditions, represented by environments $e \in \mathcal{E}$, with $|\mathcal{E}| = n_e$. For each environment $e \in \mathcal{E}$, we have access to a dataset $D^e = \{(X_i, T_i, Y_i)\}_{i=1}^n$ which contains $n$ i.i.d. tuples sampled from the marginal induced by the joint distribution $\mathbb{P}^e$ over $(X, U, T, Y)$. Here, $X \in \mathbb{R}^d$ are the observed covariates, $U \in \mathbb{R}^k$ are the unobserved covariates, $T \in \{0, 1\}$ is a treatment assignment variable and $Y \in \mathbb{R}$ is the outcome. We denote by $\overline{\mathbb{P}} = \frac{1}{|\mathcal{E}|} \sum_{e \in \mathcal{E}} \mathbb{P}^e$ the distribution of the pooled environments.

For each environment $e \in \mathcal{E}$, the distribution $\mathbb{P}^e$ is induced by a structural causal model (SCM), defined as a tuple $\mathcal{M}^e = (\mathcal{G}, \{f_i\}_{i=1}^p, \mathbb{P}^e_\epsilon)$ on $p = d + k + 2$ variables $(\tilde{Z}_1, \ldots, \tilde{Z}_p)$, where the observed covariates are $X = \tilde{Z}_{[d]}$, the unobserved covariates are $U = \tilde{Z}_{d+[k]}$, the treatment variable is $T = \tilde{Z}_{p-1}$, and the outcome variable is $Y = \tilde{Z}_p$, with $p \notin \mathrm{An}(T)$. The SCM defines the probability distribution $\mathbb{P}^e$ by setting for each $j \in [p]$

$$\tilde{Z}_j \leftarrow f_j(\tilde{Z}_{\mathrm{Pa}(\tilde{Z}_j)}, \epsilon_j), \quad j = 1, \ldots, p, \tag{1}$$

where $f_j : \mathbb{R}^p \times \mathbb{R} \to \mathbb{R}$ is a measurable function and $\epsilon \in \mathbb{R}^p$ is an exogenous noise vector following the joint distribution $\mathbb{P}^e_\epsilon$ over $p$ independent variables.

Further, along the lines of the existing methods in the literature (Shi et al., 2021; Wang et al., 2023), we require the absence of observed mediators between $T$ and $Y$ in the structural causal model.

**Assumption 3.1** (Absence of Mediators)**.** *We assume that no observed mediators exist between $T$ and $Y$, i.e. it holds that*

$$\mathrm{De}(T) \cap \mathrm{An}(Y) \cap [d] = \emptyset.$$

We remark that Assumption 3.1 is *falsifiable* using statistical tests to determine whether a covariate is a mediator between $T$ and $Y$; see, e.g. Baron & Kenny (1986); Preacher & Hayes (2004). Furthermore, even when this assumption is violated, the causal quantity identified by our method corresponds to the natural direct effect (Pearl, 2022), which remains a quantity of interest in fields such as epidemiology (Tchetgen & VanderWeele, 2014) and the social sciences (Imai et al., 2011).

## 3.1 TREATMENT EFFECT IDENTIFICATION

Our goal is to identify the treatment effects for different environments in the presence of unobserved and post-treatment variables. More specifically, we are interested in the average treatment effects (ATEs) for all environments $e \in \mathcal{E}$, defined as

$$\theta^e = \mathbb{E}_{\mathbb{P}^e}\left[Y^{\mathrm{do}(T=1)} - Y^{\mathrm{do}(T=0)}\right].$$

A common approach for identifying the ATE is to find a *valid adjustment set* (Shpitser et al., 2010), that is, a subset $S \subseteq [d]$ of the observed covariates that satisfies both the classic outcome and treatment identification formulae, i.e. for all environments $e \in \mathcal{E}$ and $t \in \{0, 1\}$ it holds that

$$\mathbb{E}_{\mathbb{P}^e}\left[Y^{\mathrm{do}(T=t)}\right] = \mathbb{E}_{\mathbb{P}^e}\left[\mathbb{E}_{\mathbb{P}^e}[Y \mid X_S, T = t]\right] = \mathbb{E}_{\mathbb{P}^e}\left[\frac{Y\mathbb{I}\{T = t\}}{\mathbb{P}^e(T = t \mid X_S)}\right]. \tag{2}$$

Several criteria have been proposed in the literature to find valid adjustment sets, with the backdoor criterion being the most prominent—see Peters et al. (2017, Sec. 6.6) for a detailed discussion. However, these criteria crucially rely on knowledge of the underlying causal graph. Therefore, it is commonly assumed among practitioners that the set $[d]$ of all observed covariates is a valid adjustment set. This is a reasonable assumption only in settings where all the observed covariates are pre-treatment and there are no unobserved covariates.

In contrast, our work focuses on settings where both post-treatment and unobserved covariates are present. To identify the ATE in such settings, we introduce a key assumption. While each environment may have a different joint distribution $\mathbb{P}^e$ over $(X, U, Y, T)$, we assume that at least one of $T$ or $Y$ is an invariant node with its parents fully observed and the conditional mean of the node given its parents is invariant across environments.

**Assumption 3.2** (Invariant node). *We assume that one of the following holds for all $e \in \mathcal{E}$:*

*(a) All parents $\mathrm{Pa}(T)$ of $T$ are observed and $\mathbb{E}_{\mathbb{P}^e}\left[T \mid Z_{\mathrm{Pa}(T)}\right] = \mathbb{E}_{\overline{\mathbb{P}}}\left[T \mid Z_{\mathrm{Pa}(T)}\right],\ \mathbb{P}^e - \text{a.s.}$*

*(b) All parents $\mathrm{Pa}(Y)$ of $Y$ are observed and $\mathbb{E}_{\mathbb{P}^e}\left[Y \mid Z_{\mathrm{Pa}(Y)}\right] = \mathbb{E}_{\overline{\mathbb{P}}}\left[Y \mid Z_{\mathrm{Pa}(Y)}\right],\ \mathbb{P}^e - \text{a.s.}$*

*We denote the node for which the above holds as the **invariant node** $V_{\mathrm{inv}} \in \{T, Y\}$.*

It is worth emphasizing that each of the above assumptions can provide identification of the ATE on its own. Here, we combine these two identification assumptions to obtain doubly robust identification: we only require that either (a) or (b) in Assumption 3.2 holds. This is similar in spirit to the double machine learning literature (Robins & Rotnitzky, 1995; Chernozhukov et al., 2018), where only one of two assumptions about model specification needs to hold to obtain consistent ATE estimates. However, the key difference is that our assumption offers robustness against potential unobserved variables in the underlying causal graph, whereas classic double robustness offers robustness against misspecification of the outcome and treatment functions.

Further, the invariance assumptions (a) and (b) are closely related to the conditions in the invariance-based domain generalization literature, such as Peters et al. (2016); Rojas-Carulla et al. (2018); Gu et al. (2024). While these settings are included in Assumption 3.2 (as we discuss in Appendix A.1), our setting does not require full independence of the noise variable[2], unlike Peters et al. (2016), nor is it limited to the additive noise case, as in Gu et al. (2024), which does not hold in the case of binary treatment variables. Finally, we comment on the observability part of Assumption 3.2: assuming $\mathrm{Pa}(V_{\mathrm{inv}})$ are observed is strictly weaker than causal sufficiency, which would require the full causal graph to be observed. Notably, our framework allows for scenarios where Assumption 3.2 (a) is satisfied with $T$ as the invariant node, whereas the setting proposed in Shi et al. (2021) is more restrictive, since it requires $Y$ to always be the invariant node.

## 4 METHODOLOGY

In this section, we introduce RAMEN, our method to identify the ATE by leveraging the heterogeneity in the observed data. First, we present a doubly robust population-level estimator and discuss

---

[2]Although we require independence of exogenous noise variables for the *full* graph, here, we refer to the graph limited to the observed nodes, where the noise variables $\epsilon_j$ can be dependent on $Z_{\mathrm{Pa}(Z_j)}$.

under which conditions it equals to the ATE. Then, we show how to compute this estimator tractably by minimizing a novel invariance loss and propose two algorithms to do so: a combinatorial search over subsets and a more scalable differentiable approach for high-dimensional covariate settings.

## 4.1 POPULATION-LEVEL ESTIMATOR

In what follows, we denote by $\mathcal{I}_T = [d]$ and $\mathcal{I}_Y = [d+1]$ the index sets corresponding to the observed variables $Z = (X, T)$. For any node $V \in \{T, Y\}$ and any observed subset $S \subseteq \mathcal{I}_V$, we define the conditional means over the pooled and individual environments

$$\overline{m}_S(Z; V) := \mathbb{E}_{\overline{\mathbb{P}}}[V \mid Z_S] \quad \text{and} \quad m_S^e(Z; V) := \mathbb{E}_{\mathbb{P}^e}[V \mid Z_S].$$

**1. Identify an invariant set** We begin by observing that, by Assumption 3.2, there exists an invariant node $V_{\text{inv}} \in \{T, Y\}$ and a subset of covariates $S$ (given by, e.g, $\text{Pa}(V_{\text{inv}})$), for which the following conditional moment constraint holds for all environments $e \in \mathcal{E}$:

$$\exists S \subseteq \mathcal{I}_{V_{\text{inv}}} : \quad m_S^e(Z; V_{\text{inv}}) = \overline{m}_S(Z; V_{\text{inv}}), \quad \mathbb{P}^e - \text{a.s.} \tag{3}$$

The set $S$ is not necessarily unique: besides the (observed) parents of $V_{\text{inv}}$ for instance, the invariance could also hold for certain supersets of $\text{Pa}(V_{\text{inv}})$. By observing that the conditional moment constraint above is equivalent to the following infinite set of unconditional moment constraints

$$\mathbb{E}_{\mathbb{P}^e}\left[(V_{\text{inv}} - \overline{m}_S(Z; V_{\text{inv}}))h(Z_S)\right] = 0, \quad \text{for all measurable } h,$$

any set $S$ that satisfies the invariance constraint Equation (3) is also contained in

$$\underset{S \subseteq \mathcal{I}_{V_{\text{inv}}}}{\text{argmin}} \max_{e \in \mathcal{E}} \left( \sup_{h \in L^0\left(\mathbb{R}^{|S|}\right)} \mathbb{E}_{\mathbb{P}^e}\left[(V_{\text{inv}} - \overline{m}_S(Z; V_{\text{inv}}))h(Z_S)\right] \right)^2 := \underset{S \subseteq \mathcal{I}_{V_{\text{inv}}}}{\text{argmin}} \; J_S(Z; V_{\text{inv}}), \tag{4}$$

where $L^0(\mathbb{R}^d)$ denotes the space of measurable functions over $\mathbb{R}^d$. However, since the invariant node $V_{\text{inv}}$ is not known beforehand, we search for a set of observed nodes that satisfy the invariance with respect to either $T$ or $Y$, that is, we want to find

$$S_{\text{⚲}} \in \min_{V \in \{T, Y\}} \underset{S \subseteq \mathcal{I}_V}{\text{argmin}} \quad J_S(Z; V). \tag{5}$$

Further, we can leverage the structural knowledge that $T$ is always an ancestor of $Y$ to simplify the optimization problem. Specifically, we know that $T$ must be part of the invariant set when $V = Y$. Therefore, we can condition the expectation in Equation (4) on $T = 0$ and $T = 1$ separately and take the maximum of the resulting losses. This gives us a slightly different loss function for $Y$:

$$J_S(Z; Y) := \max_{t \in \{0,1\}} \max_{e \in \mathcal{E}} \left( \sup_{h \in L^0\left(\mathbb{R}^{|S|}\right)} \mathbb{E}_{\mathbb{P}^e}\left[\left(Y - \mathbb{E}_{\overline{\mathbb{P}}}[Y \mid Z_S, T = t]\right) h(Z_S) \mid T = t\right] \right)^2. \tag{6}$$

**2. Estimate the ATE** For a minimizer $S_{\text{⚲}}$, we then define the corresponding population-level RAMEN, estimator for all environments $e \in \mathcal{E}$ as

$$\theta^e(S_{\text{⚲}}) := \mathbb{E}_{\mathbb{P}^e}\left[\bar{\mu}_1(X_{S_{\text{⚲}}}) - \bar{\mu}_0(X_{S_{\text{⚲}}}) + \frac{(Y - \bar{\mu}_1(X_{S_{\text{⚲}}}))T}{\bar{\pi}(X_{S_{\text{⚲}}})} - \frac{(Y - \bar{\mu}_0(X_{S_{\text{⚲}}}))(1 - T)}{1 - \bar{\pi}(X_{S_{\text{⚲}}})}\right],$$

where we define the pooled conditional outcome and treatment functions as

$$\bar{\mu}_t(X_{S_{\text{⚲}}}) := \mathbb{E}_{\overline{\mathbb{P}}}[Y \mid X_{S_{\text{⚲}}}, T = t] \quad \text{and} \quad \bar{\pi}(X_{S_{\text{⚲}}}) := \mathbb{E}_{\overline{\mathbb{P}}}[T \mid X_{S_{\text{⚲}}}].$$

In what follows, we show that under a condition on data heterogeneity, detailed in Assumption 4.1, our population-level estimator $\theta^e(S_{\text{⚲}})$ is equivalent for all $S_{\text{⚲}}$ that satisfy Equation (5), and it is equal to the true treatment effect. In Section 4.3, we then discuss how we can construct a good finite-sample ATE estimate in a computationally efficient way.

### 4.2 DOUBLY ROBUST IDENTIFICATION GUARANTEES

Without further assumptions, finding the minimizer of Equation (5) is not sufficient for identifying the ATE: for instance, if there is no variability between distributions $\mathbb{P}^e$, our objective could be trivially minimized by any observed subset $S$. Only when there is sufficient heterogeneity in the observed environments will $\theta^e(S_{\oplus})$ be equivalent to the ATE. We formalize this condition below.

**Assumption 4.1** (Identification condition). *For all $V \in \{T, Y\}$ and $S \subseteq \mathcal{I}_V$, it holds that:*

$$\forall e \in \mathcal{E} : m_S^e(Z; V) = \overline{m}_S(Z; V), \quad \mathbb{P}^e-\text{a.s.} \implies \overline{\mathbb{P}}\left(\overline{m}_S(Z; V) \neq \overline{m}_{\text{Pa}(V)}(Z; V)\right) = 0.$$

Assumption 4.1 can be understood as ensuring that the environments present sufficient heterogeneity. This heterogeneity is crucial because it guarantees that conditioning on any set $S$ with invariant outcome or treatment functions across environments is equivalent to conditioning on the parents of the invariant node. Although our environment heterogeneity assumption is relatively strict, it is a common requirement in the invariance literature (cf. Peters et al. (2016); Arjovsky et al. (2019)). For example, in the simultaneous noise intervention setting described in Peters et al. (2016, Section 4.2.3), Assumption 4.1 can be satisfied with as few as two environments, while in the case of single-node interventions it requires $\mathcal{O}(p)$ environments, where $p$ is the number of observed variables.

We now present our formal identification result for the ATE.

**Theorem 1** (Doubly robust identification). *Let $S_{\oplus}$ be any minimizer of the invariance loss in Equation (5). Then, under Assumptions 3.1, 3.2, 4.1, if positivity holds, that is*

$$\forall e \in \mathcal{E} : \quad \mathbb{P}^e(T = t \mid X_{S_{\oplus}} = x) > 0, \ \forall t \in \{0, 1\} \text{ and } \forall x \in \text{supp}\left(\mathbb{P}_X^e\right),$$

*we can identify the average treatment effect $\theta^e = \theta^e(S_{\oplus})$, for all environments $e \in \mathcal{E}$.*

We remark that the positivity assumption is standard and widely used for identifying treatment effects in observational studies (Hernán & Robins, 2010, Sec. 3.2). Theorem 1 states that any solution to our invariance loss is a valid adjustment set in the sense that it is sufficient to identify the average treatment effect in all the environments.

### 4.3 AN EFFICIENT FINITE-SAMPLE ESTIMATOR

The population-level estimator presented above involves two significant computational challenges. First, the invariance loss involves a supremum over an infinite-dimensional space of measurable functions, making it intractable to compute directly. Second, it requires searching over all possible subsets of covariates, which is computationally infeasible for high-dimensional settings. To address these issues, we introduce a practical estimator based on a kernelized invariance loss and a differentiable relaxation of the subset selection problem.

**Kernelized invariance loss** A major problem of the loss function in Equation (4) is that it is computationally infeasible to search over the entire space of measurable functions. However, we can simplify the problem by restricting $h$ to be in a reproducing kernel Hilbert space (RKHS). As long as the reproducing kernel of the RKHS is universal (e.g. Gaussian kernel), the two formulations are equivalent (Gretton et al., 2012). More formally, for any subset $S \subseteq \mathcal{I}_V$ and environment $e \in \mathcal{E}$:

$$\left(\sup_{h \in L^0\left(\mathbb{R}^{|S|}\right)} \mathbb{E}_{\mathbb{P}^e}\left[\underbrace{(V - \overline{m}_S(Z; V))}_{:=\delta_S(Z, V)} h(Z_S)\right]\right)^2 = \left(\sup_{\|h\|_{\mathcal{H}} \leq 1} \mathbb{E}_{\mathbb{P}^e}\left[\delta_S(Z, V) h(Z_S)\right]\right)^2$$

$$= \left\|\mathbb{E}_{\mathbb{P}^e}\left[\delta_S(Z, V) k(\cdot, Z_S)\right]\right\|_{\mathcal{H}}^2$$

$$= \mathbb{E}_{\mathbb{P}^e}\left[\delta_S(Z, V) k\left(Z_S, Z_S'\right) \delta_S(Z', V')\right],$$

where $k$ is a uniformly bounded reproducing kernel corresponding to a universal RKHS $\mathcal{H}$ (Steinwart, 2001, Definition 4), and $(V', Z')$ is an independent copy of $(V, Z)$ following the same distribution. Hence, we can rewrite our invariance loss in closed form:

$$J_S(Z; V) = \max_{e \in \mathcal{E}} \mathbb{E}_{\mathbb{P}^e}\left[\delta_S(Z, V) k\left(Z_S, Z_S'\right) \delta_S(Z', V')\right]. \tag{7}$$

A long line of work has proposed methods to estimate invariant predictors, especially when the optimal predictor is linear. These methods broadly fall into two categories: hypothesis test-based methods (Peters et al., 2016; Heinze-Deml et al., 2018; Pfister et al., 2019) and optimization-based methods (Arjovsky et al., 2019; Ghassami et al., 2017; Rothenhäusler et al., 2019; 2021; Pfister et al., 2021; Yin et al., 2024; Shen et al., 2023; Gu et al., 2024; Wang et al., 2024). Our approach falls in the latter category, with a fundamental distinction. While all these works utilize the invariance principle to improve prediction in unseen environments and generalize to new settings, we aim to identify a treatment effect within the observed environments. This is reflected in our loss function, as it does not measure the quality of the predictor (e.g. using a least squares loss). Nonetheless, our invariance loss could also be of interest in the domain generalization literature as it retains the benefits of the invariance loss in Gu et al. (2024) while significantly simplifying their optimization procedure.

**A fully differentiable loss** When searching over all possible subsets of covariates is computationally infeasible, we propose a continuous relaxation of the optimization problem in Equation (5) that can be efficiently solved using gradient descent. Specifically, we select the nodes as $Z_w := B(w) \odot Z$, where $Z = (X, T)$ and the $j$-th component of $B(w) \in \{0,1\}^{d+1}$ is sampled independently from a Bernoulli distribution with probability $\mathrm{sigmoid}(w_j)$. We parametrize the conditional mean using a neural network $f_\theta$ and we aim to solve the following optimization problem:

$$w^{\circlearrowleft} \in \underset{w \in \mathbb{R}^{d+1}}{\mathrm{argmin}} \; \underset{\theta \in \mathbb{R}^{d+1}, V \in \{T,Y\}}{\min} \; \underset{e \in \mathcal{E}}{\max} \; \mathbb{E}_{\mathbb{P}^e, B(w)} \left[ (V - f_\theta(Z_w)) k\left(Z_w, Z_w'\right) (V' - f_\theta(Z_w')) \right],$$

Since the weights are discrete, direct differentiation is not possible. To overcome this, we use a Gumbel approximation (Jang et al., 2017; Maddison et al., 2017; Gu et al., 2024), where the $j$-th component of $B(w)$ is approximated as:

$$B_j(w) \approx \mathrm{sigmoid}\left( \frac{w_j + G_{1,j} - G_{2,j}}{\tau} \right), \quad \text{as} \quad \tau \to 0^+,$$

with $G_{1,j}$ and $G_{2,j}$ being $\mathrm{Gumbel}(0,1)$ random variables. This approximation makes $B(w)$ differentiable (where it was previously discontinuous in $w_j$), allowing us to optimize using gradient descent while gradually annealing the hyperparameter $\tau$. Finally, we construct the subset of covariates $S_{\mathrm{insta}-\circlearrowleft}$ by including $Z_i$ only if the weights are positive, that is $S_{\mathrm{insta}-\circlearrowleft} = \{i : w_i^{\circlearrowleft} > 0\}$. We refer the reader to Appendix A.3 for the complete implementation details of our algorithms.

## 5 EXPERIMENTS

In this section, we evaluate our method through experiments on synthetic, semi-synthetic, and real-world datasets. We first present experiments on several known DAGs, where the invariances are known and satisfy our assumptions. In line with our theory, RAMEN correctly identifies the ATE, resulting in a low estimation error, whereas other methods tend to fail. We also test RAMEN on a more challenging benchmark by uniformly sampling DAGs using the Erdős–Rényi model—a standard approach for testing causal methods across a wide variety of graph topologies (Huang et al., 2020). Finally, we validate our estimator beyond purely synthetic data: first in a semi-synthetic setting with real-world covariates and then in a real-world setting where we compare the conclusions from RAMEN with established epidemiological findings.

In our experiments, we focus on the statistical task of estimating the average treatment effect (ATE) $\theta^e$ for each environment $e \in \mathcal{E}$. To evaluate the performance of an estimator $\hat{\theta}^e$, we compute the mean absolute error (MAE) averaged across environments: $\frac{1}{|\mathcal{E}|} \sum_{e \in \mathcal{E}} |\theta^e - \hat{\theta}^e|$. We evaluate two implementations of RAMEN: (i) $\hat{\theta}_{\circlearrowleft} = \hat{\theta}(S_{\circlearrowleft})$, based on combinatorial subset search (Section 4.3), and (ii) $\hat{\theta}_{\mathrm{insta}-\circlearrowleft} = \hat{\theta}(S_{\mathrm{insta}-\circlearrowleft})$, based on the Gumbel trick (Section 4.3); see Appendix A.3 for the complete implementation details of our algorithms. We compare RAMEN against three baselines: $\hat{\theta}_{\mathrm{irm}}$, the IRM approach for treatment effect estimation proposed by Shi et al. (2021); $\hat{\theta}_{\mathrm{all}}$, which adjusts for all available covariates; and $\hat{\theta}_{\mathrm{null}}$, which does not adjust for any covariates.

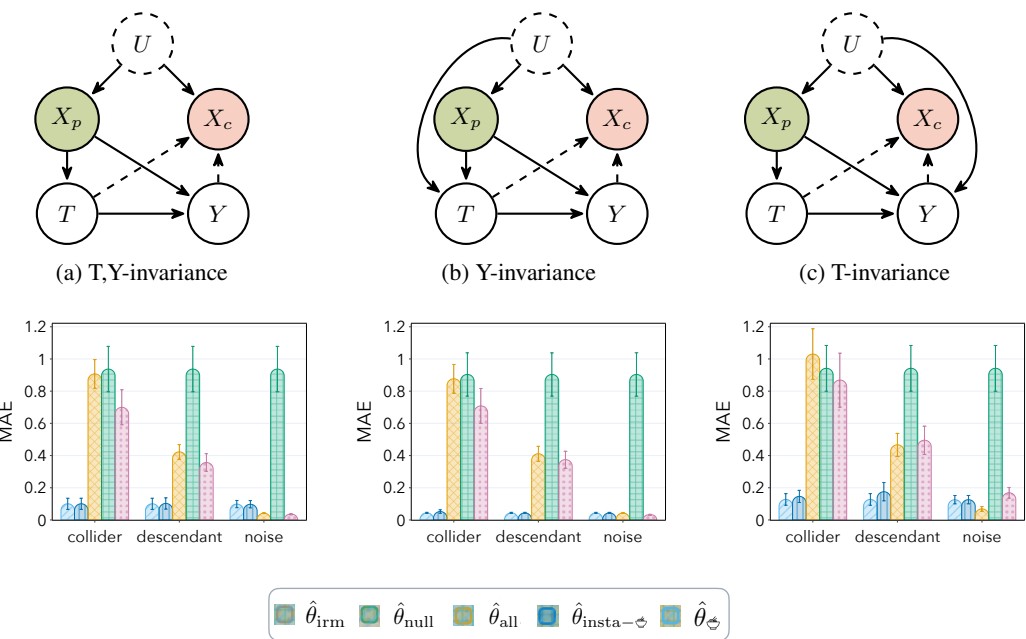

Figure 2: **(Top row)** Graphical models illustrating three scenarios in which the unobserved confounder $U$ may break different invariances between $T$, $Y$, and the observed covariates $X$. In each graph, dashed arrows denote optional edges, dashed nodes indicate unobserved variables, green and red nodes indicate good and bad controls, respectively. Panel **(a)** shows the case where $U$ does not break any invariance. Panel **(b)** shows the case where $U$ breaks the invariance between $X_p$ and $T$. Panel **(c)** shows the case where $U$ breaks the invariance between $X_p$ and $Y$. **(Bottom row)** Mean absolute errors (MAE) of four estimators are shown across 5 environments ($n = 2500$, $d = 5$ observed covariates), with error bars representing standard errors over 20 runs.

## 5.1 SYNTHETIC EXPERIMENTS WITH KNOWN DAGS

We start with data generated from distributions with simple underlying DAGs that satisfy our invariance assumptions, as illustrated in Figure 2 (Row 2). Most importantly, we consider three distinct scenarios[3]: (a) Y and T-invariances, i.e. both (a) and (b) in Assumption 3.2 hold; (b) Y-invariance, i.e. only Assumption 3.2 (b) holds; (c) T-invariance, i.e. only Assumption 3.2 (a) holds. For each of the three different invariance scenarios, we further consider three variants: where $X_c$ is either a descendant of $Y$, a collider between $T$ and $Y$, or independent noise. We describe the complete data-generating process in Appendix D.1. In the infinite sample limit, $\hat{\theta}_{\text{null}}$ should generally be biased since there is a confounder between $T$ and $Y$; $\hat{\theta}_{\text{all}}$ should be biased only when $X_c$ is a collider or a descendant; $\hat{\theta}_{\text{irm}}$ should be biased in the T-invariance case; $\hat{\theta}_{\circlearrowleft}$ and $\hat{\theta}_{\text{insta}-\circlearrowleft}$ should never be biased.

In Figure 2 (Row 1), we present the empirical MAE for all methods on finite-sample experiments that confirm the predictions from theory. First of all, both of our methods, $\hat{\theta}_{\circlearrowleft}$ and $\hat{\theta}_{\text{insta}-\circlearrowleft}$, consistently achieve lower MAE compared to the baselines in all scenarios. In particular, we observe that the differentiable relaxation of our method does not significantly compromise statistical performance. Further, for T-invariance, the performance of $\hat{\theta}_{\text{irm}}$ deteriorates markedly as expected —e.g. in scenarios where the post-treatment variable is a descendant of $Y$, it performs worse than simply adjusting for all available covariates. In contrast, our approach remains robust even when one of the invariances is compromised. Finally, we observe that relying on T-invariance increases the error across methods, possibly because the adjustment set we recover, the parents of the treatment, leads to a statistically less efficient estimator, see e.g. Henckel et al. (2022, Corollary 3.4).

---

[3]In Appendix C.2, we also present experiments for cases when none of the invariances hold.

## 5.2 Synthetic experiment with random high dimensional DAGs

We randomly draw a graph from the Erdös-Rényi random graph model with a total number of nodes $p = 20$. We do rejection sampling to exclude graphs that either contain mediators—as they violate Assumption 3.1—or do not contain at least a confounder. We then assign $Y$ and $T$ to nodes such that the invariance assumption is satisfied for at least one of them[4], and sample all variables from the resulting DAG via a linear structural causal model except for the treatment variable $T$. We sample $T$ from a Bernoulli distribution with parameter equal to the sigmoid function applied to the function $f$ in the structural equation of $T$ as in Equation (1). We further post-process the graph by adding a node $X_c = Y + T$ to make sure that there is at least one post-treatment covariate. We assign the parents of $T$ or $Y$ (except common parents) to be the set of unobserved covariates $U$, depending on the invariance we want to preserve.

In each environment, we apply a random uniform mean and variance shift to all the nodes in the graph except for $T$ and $Y$ while preserving Assumption 4.1; see Appendix D.1 for further details on the data generation. This process leads to distributions that are guaranteed to satisfy all our assumptions in Theorem 1.

In our experiments we sample 100 DAGs and for each DAG vary the number of environments while keeping the sample size fixed. In Figure 3, we plot the empirical MAE of $\hat{\theta}_{\text{insta}-\circlearrowleft}$ ($\hat{\theta}_{\circlearrowleft}$ is computationally infeasible) and the baseline estimators averaged across the DAGs as a function of the number of environments. Notably, we observe that across all settings and numbers of available environments, $\hat{\theta}_{\text{insta}-\circlearrowleft}$ significantly outperforms all the other baselines. Expectedly, $\hat{\theta}_{\text{irm}}$ fails to surpass all trivial baselines, even with many environments, as it lacks the robustness to unobserved parents of $Y$.

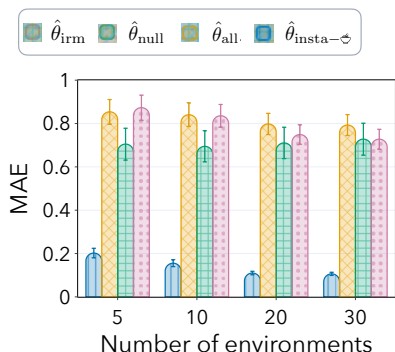

Figure 3: We plot the mean absolute error averaged across environments when the $T$-invariance is preserved. We sample $n = 2000$ points for each environment; we report mean and standard error over 100 runs.

## 5.3 Semi-synthetic experiments: IHDP

The IHDP dataset contains covariates from $n = 748$ low-birth-weight, premature infants enrolled in a home visitation program designed to improve their cognitive scores (Hill, 2011). Instead of using the commonly adopted synthetic functions from Dorie (2016), we simulate a more challenging non-linear version of the outcome and treatment mechanisms inspired by Kang & Schafer (2007). Specifically, we retain the 6 continuous covariates (out of 25 total covariates) from the original dataset and simulate the outcome $Y$ and treatment assignment $T$ by randomly sampling complex functional forms, such as exponentials and polynomials. In addition, we introduce a 2-dimensional synthetic collider, $X_c$, as a linear function of $T$ and $Y$. We generate environments using Gaussian mean shifts in both pre-and post-treatment features, as well as in *either* $Y$ or $T$ or neither of them, and set the number of environments to $|\mathcal{E}| = 5$. Finally, at inference time we do not observe one of the parents of the node among $Y, T$ that is not invariant. See Appendix D.2 for the complete experimental details.

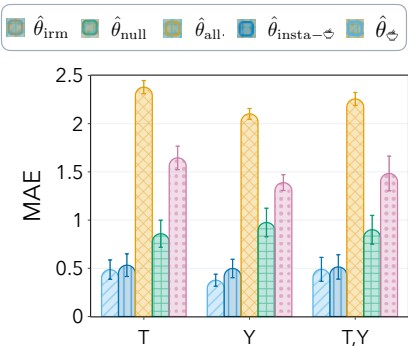

Figure 4: Mean absolute error averaged across environments for the IHDP dataset when different invariances are preserved (T, Y, or both). We consider five environments with $n = 748$ points each; mean and standard error are reported over 20 runs.

---

[4]In the main text we present the results for the settings where the parents $Y$, excluding the parents of $T$, are unobserved. Please refer to Appendix C.3 for additional experiments where other invariances hold

Figure 4 depicts the MAE of all the methods when different invariance assumptions hold. Similar to the synthetic experiments in Sections 5.1 and 5.2, $\hat{\theta}_{\text{irm}}$ exhibits higher MAE when $Y$ is not invariant across environments, and $\hat{\theta}_{\text{all}}$, adjusting for all features, generally results in poor performance. Interestingly, $\hat{\theta}_{\text{null}}$ performs competitively since the confounders have a limited impact on the outcome and treatment assignment in this dataset. Additional experiments where the post-treatment feature is either a descendant of the outcome, independent noise, or where neither $T$ nor $Y$ remains invariant are provided in Appendix C.2. Moreover, we present experiments including mediators between the treatment and the outcome in Appendix C.1.

## 5.4 REAL-WORLD EXPERIMENT: EFFECT OF MATERNAL SMOKING ON BIRTH WEIGHT

In our real-world experiment, we evaluate our method on the observational dataset from Cattaneo (2010) that studies the effect of maternal smoking (treatment $T$) during pregnancy on birth weight (outcome $Y$) using the data from $n = 4642$ patients. We split the original dataset into $|\mathcal{E}| = 4$ environments defined by the trimester of birth. We then use 20 other variables from the original dataset as observed covariates $X$. Given the nature of the treatment, we expect that some features are post-treatment, i.e. measured after the mother started smoking, as noted in Wilcox (2001). We provide complete experimental details in Appendix D.3.

Table 1: ATE estimates for the Cattaneo2 dataset using different baselines. We report the mean and standard deviation over 100 initializations of the random seeds in the algorithms.

| Method | ATE (mean $\pm$ std) |
|---|---|
| $\hat{\theta}_{\text{null}}$ | $-275.25 \pm 10^{-5}$ |
| $\hat{\theta}_{\text{all}}$ | $-157.55 \pm 10^{-5}$ |
| $\hat{\theta}_{\text{irm}}$ | $-182.65 \pm 48.32$ |
| $\hat{\theta}_{\text{insta}-\circlearrowleft}$ | $-214.60 \pm 25.20$ |

Table 1 presents the results of the differentiable version of our method, alongside various baselines. While the ground truth ATE is unknown, the effect estimated by adjusting for the set selected by $\hat{\theta}_{\text{insta}-\circlearrowleft}$ aligns with existing epidemiological literature: both observational and interventional studies (Meyer & Comstock, 1972; Sexton & Hebel, 1984) as well as statistical analyses (Almond et al., 2005; Cattaneo, 2010) estimate a decrease in birth weight that ranges from 200 to 250 grams for infants born to smoking mothers compared to non-smoking mothers. In contrast, $\hat{\theta}_{\text{null}}$ overestimates the ATE, whereas both $\hat{\theta}_{\text{all}}$ and $\hat{\theta}_{\text{irm}}$ underestimate it.

## 6 DISCUSSION AND FUTURE WORK

In this work, we proposed **R**obust **A**TE identification from **M**ultiple **EN**vironments (RAMEN), a method that leverages multiple environments to identify the ATE in the presence of post-treatment and unobserved variables. To the best of our knowledge, we present the first ATE identification guarantees in this highly relevant, but previously unexplored setting. Further, we introduce a new version of double robustness that concerns unobserved variables rather than model misspecification.

Nevertheless, our method faces several limitations. First, similar to other kernel-based methods, our approach suffers from the curse of dimensionality and the computational complexity associated with computing kernel matrix. Additionally, the requirement for sufficient heterogeneity across environments may be too stringent in some practical cases. Finally, the combinatorial subset is computationally demanding, and the Gumbel trick remains a heuristic solution. Addressing any of these shortcomings would constitute interesting avenues for future work.

## ACKNOWLEDGMENTS

We thank Jonas Peters for the helpful discussions on an earlier version of this work. PDB was supported by the Hasler Foundation grant number 21050. JK was supported by the SNF grant number 204439. JA was supported by the ETH AI Center. YW was supported in part by the Office of Naval Research under grant number N00014-23-1-2590, the National Science Foundation under

Grant No. 2231174, No. 2310831, No. 2428059, No. 2435696, No. 2440954, and a Michigan Institute for Data Science Propelling Original Data Science (PODS) grant. This work was done in part while JK and FY were visiting the Simons Institute for the Theory of Computing.

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

APPENDICES

The following appendices provide deferred proofs, experiment details, and ablation studies.

**TABLE OF CONTENTS**

## A    METHODOLOGY

### A.1    DISCUSSION OF ASSUMPTION 3.2

First, we observe here that Assumption 3.2 is not a minimal "observability" condition on the parents of $Y$ and $T$: in some cases, it might still be possible to find a valid adjustment set via the observed parents of either $T$ or $Y$ (or both), although no full set of parents was observed (see e.g. Figure 5). However, in such cases, the valid adjustment set or the corresponding regression function cannot be recovered via invariance methods, since neither $T$ nor $Y$ are invariant across environments. Thus, in a way, Assumption 3.2 is a minimal assumption on the DAG if one wants to recover the ATE via invariance of conditional expectations.

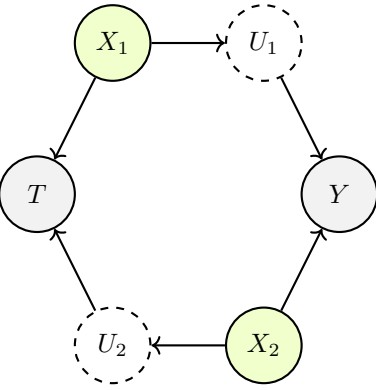

Further, we remark that our assumption is neither stronger nor weaker than the commonly used *ignorability* assumption with respect to $X$ (Robins & Greenland, 1992). For example, if parents of both $T$ and $Y$ are unobserved, Assumption 3.2 will not hold, but ignorability could still apply if no *common* parent of $T$ and $Y$ is unobserved. Conversely, in graphs with M-bias structures and colliders, the ignorability assumption will not hold.

Figure 5: Although neither full set of parents is observed, one can still find a valid adjustment set $\{X_1, X_2\}$ (in green).

Below, we give some examples of settings in which Assumption 3.2 holds, including scenarios in the existing invariance literature (Peters et al., 2016; Gu et al., 2024):

**Fully observed DAG**    In Peters et al. (2016), we are given multi-environment data $\{\mathbb{P}^e : e \in \mathcal{E}\}$, where each distribution $\mathbb{P}^e$ is induced by an SCM $\mathcal{M}^e = (\mathcal{G}, \{f_i^e\}_{i=1}^p, \mathbb{P}_\epsilon^e)$, all variables $(X_1, ..., X_d, Y)$ are observed, and for all $\mathbb{P}^e \in \mathcal{E}$ it is satisfied that

$$Y^e = g(X_{\text{Pa}(Y)}^e, \epsilon^e), \epsilon^e \sim F_\epsilon \text{ and } \epsilon^e \perp\!\!\!\perp X_{\text{Pa}(Y)}^e.$$

In particular, the independence condition implies equality of conditional distributions $\mathbb{P}^e(Y | X_{\text{Pa}(Y)}^e)$ across environments and thus Assumption 3.2(b).

**General DAG with additive noise**    In Gu et al. (2024), the target variable follows the following data generating process for all $e \in \mathcal{E}$:

$$Y^e = g(X_{S^\star}^e) + \epsilon^e; \quad \mathbb{E}[\epsilon^e | X_{S^\star}^e] = 0,$$

where $S^\star$ is the "true important variable set". This setting is, in a way, more general than Peters et al. (2016), since the noise variable is not required to be independent of the parent variables—instead, the only condition is on the first conditional moment of $\epsilon^e$. Due to the additivity of the noise, Assumption 3.2(b) follows immediately.

**General DAG with multiplicative noise**    We can define a similar setting for multiplicative noise, setting for all $e \in \mathcal{E}$:

$$Y^e = g(X_{S^\star}^e)\epsilon^e; \quad \mathbb{E}[\epsilon^e | X_{S^\star}^e] = c,$$

where $S^\star$ is, again, the true parent/important variable set, and $c$ is independent of the environment. We observe that Assumption 3.2(b) follows since it holds that

$$\mathbb{E}^e[Y | X_{S^\star}] = \mathbb{E}^e[g(X_{S^\star})\epsilon | X_{S^\star}] = g(X_{S^\star})\mathbb{E}^e[\epsilon | X_{S^\star}] = cg(X_{S^\star}).$$

**General DAG with polynomial noise**    From the above two examples, it becomes clear that for any $Y = g(X_{S^\star})p_k(\epsilon)$, where $p$ is a polynomial of degree $k$, we have that Assumption 3.2(b) holds if for all $e \in \mathcal{E}$ it holds that

$$\mathbb{E}^e[\epsilon^{k'} \mid X_{S^\star}] = c_l, \text{ for all } k' \leq k.$$

where $S^\star$ is the important variable/parent set. This condition is strictly weaker than the independence condition since $k$ is finite.

### A.2 PROOF OF THEOREM 1

First, we establish that the loss function in Equation (4) attains a value of zero at any minimizer $S_{\clubsuit}$. By Assumption 3.2, there exists an invariant node $V \in \{T, Y\}$ whose parents are observed. Since it holds that $J_{\mathrm{Pa}(V)}(Z; V) = 0$ and $\mathrm{Pa}(V) \subseteq \mathcal{I}_V$, we conclude that there exists a subset $S \subseteq \mathcal{I}_V$ such that $\min\{J_S(Z; T), J_S(Z; Y)\} = 0$. Additionally, since the loss function is non-negative, any global minimizer of Equation (4) must have a corresponding loss value of zero.

Recall that the RAMEN estimator is given by

$$\theta^e(S_{\clubsuit}) := \underbrace{\mathbb{E}_{\mathbb{P}^e}\left[\bar{\mu}_1(X_{S_{\clubsuit}}) + \frac{(Y - \bar{\mu}_1(X_{S_{\clubsuit}}))T}{\bar{\pi}(X_{S_{\clubsuit}})}\right]}_{T_1} - \underbrace{\mathbb{E}_{\mathbb{P}^e}\left[\bar{\mu}_0(X_{S_{\clubsuit}}) + \frac{(Y - \bar{\mu}_0(X_{S_{\clubsuit}}))(1 - T)}{1 - \bar{\pi}(X_{S_{\clubsuit}})}\right]}_{T_2}.$$

where we have slightly rearranged the order of the terms. In the following, we only consider the treated term $T_1$ and prove that $T_1 = \mathbb{E}_{\mathbb{P}^e}\left[Y^{\mathrm{do}(T=1)}\right]$. Analogous reasoning for the control term $T_2$ shows that $T_2 = \mathbb{E}_{\mathbb{P}^e}\left[Y^{\mathrm{do}(T=0)}\right]$ and thus proves the claim.

We now consider two cases, depending on whether the minimum is attained for the node $Y$ or $T$.

**Case 1:** $J_{S_{\clubsuit}}(Z; T) = 0$. Since we assume that the kernel belongs to a universal RKHS (Steinwart, 2001, Def. 4), it follows from Gretton et al. (2012, Theorem 5) that for all $e \in \mathcal{E}$

$$\overline{m}_{S_{\clubsuit}}^e(Z; T) = \mathbb{E}_{\mathbb{P}^e}[T \mid X_{S_{\clubsuit}}] = \bar{\pi}\left(X_{S_{\clubsuit}}\right), \ \mathbb{P}^e - \text{a.s.} \tag{8}$$

Then, by Assumption 4.1, it holds that

$$\bar{\pi}(X_{S_{\clubsuit}}) = \bar{\pi}(Z_{\mathrm{Pa}(T)}), \ \overline{\mathbb{P}} - \text{a.s.}$$

Further, since the measure $\overline{\mathbb{P}}$ dominates $\mathbb{P}^e$, it holds that

$$\bar{\pi}(X_{S_{\clubsuit}}) = \bar{\pi}(Z_{\mathrm{Pa}(T)}), \ \mathbb{P}^e - \text{a.s.} \tag{9}$$

Using the tower property, we compute

$$\mathbb{E}_{\mathbb{P}^e}\left[\left(1 - \frac{T}{\bar{\pi}(X_{S_{\clubsuit}})}\right)\bar{\mu}_1(X_{S_{\clubsuit}})\right] = \mathbb{E}_{\mathbb{P}^e}\left[\bar{\mu}_1(X_{S_{\clubsuit}})\right] - \mathbb{E}_{\mathbb{P}^e}\left[\mathbb{E}_{\mathbb{P}^e}\left[\frac{T}{\bar{\pi}(X_{S_{\clubsuit}})} \mid X_{S_{\clubsuit}}\right]\bar{\mu}_1(X_{S_{\clubsuit}})\right]$$

$$= \mathbb{E}_{\mathbb{P}^e}\left[\bar{\mu}_1(X_{S_{\clubsuit}})\right] - \mathbb{E}_{\mathbb{P}^e}\left[\bar{\mu}_1(X_{S_{\clubsuit}})\right] = 0.$$

Hence,

$$T_1 = \mathbb{E}_{\mathbb{P}^e}\left[\frac{TY}{\bar{\pi}(X_{S_{\clubsuit}})}\right] = \mathbb{E}_{\mathbb{P}^e}\left[\frac{TY}{\bar{\pi}(Z_{\mathrm{Pa}(T)})}\right],$$

where the second equality follows by (9).

Now, under the positivity assumption and Assumption 3.2, since the parents of $T$ satisfy the backdoor criteria, we can identify the treatment effect, that is, it holds that

$$\mathbb{E}_{\mathbb{P}^e}\left[\frac{TY}{\bar{\pi}(Z_{\mathrm{Pa}(T)})}\right] = \mathbb{E}_{\mathbb{P}^e}\left[Y^{\mathrm{do}(T=1)}\right].$$

Thus, we conclude that

$$\mathbb{E}_{\mathbb{P}^e}\left[Y^{\mathrm{do}(T=1)}\right] = \mathbb{E}_{\mathbb{P}^e}\left[\frac{TY}{\bar{\pi}(X_{S_{\clubsuit}})} + \left(1 - \frac{T}{\bar{\pi}(X_{S_{\clubsuit}})}\right)\bar{\mu}_1(X_{S_{\clubsuit}})\right].$$

The analogous argument can be used to show $T_2 = \mathbb{E}_{\mathbb{P}^e}\left[Y^{\mathrm{do}(T=0)}\right]$. Combining the terms yields

$$\theta^e(S_{\clubsuit}) = \mathbb{E}_{\mathbb{P}^e}\left[Y^{\mathrm{do}(T=1)}\right] - \mathbb{E}_{\mathbb{P}^e}\left[Y^{\mathrm{do}(T=0)}\right].$$

**Case 2:** $J_{S_{\cancel{Y}}}(Z; Y) = 0$. We recall that this term of the loss function is defined as

$$J_S(Z; Y) := \max_{t \in \{0,1\}} \max_{e \in \mathcal{E}} \left( \sup_{h \in L^0(\mathbb{R}^{|S|})} \mathbb{E}_{\mathbb{P}^e} \left[ \left( Y - \mathbb{E}_{\overline{\mathbb{P}}}[Y \mid Z_S, T = t] \right) h(Z_S) \mid T = t \right] \right)^2.$$

Thus, again, by the universal property of the kernel $k$ and Gretton et al. (2012, Theorem 5), it follows

$$\forall e \in \mathcal{E}: \quad \mathbb{E}_{\mathbb{P}^e}[Y \mid T = t, X_{S_{\cancel{Y}}}] = \bar{\mu}_t \left( X_{S_{\cancel{Y}}} \right), \quad \mathbb{P}^e - \text{a.s.}, \ \forall t \in \{0,1\}. \tag{10}$$

Further, by Assumption 4.1, we have

$$\bar{\mu}_t(X_{S_{\cancel{Y}}}) = \bar{\mu}_t(Z_{\mathrm{Pa}(Y)}), \quad \overline{\mathbb{P}} - \text{a.s.}$$

This also implies $\mathbb{P}^e - \text{a.s.}$ equality of the conditional means for all $e \in \mathcal{E}$ by domination of measure. We now observe that

$$\mathbb{E}_{\mathbb{P}^e} \left[ \bar{\mu}_1(X_{S_{\cancel{Y}}}) \right] = \mathbb{E}_{\mathbb{P}^e} \left[ \bar{\mu}_1(Z_{\mathrm{Pa}(Y)}) \right] = \mathbb{E}_{\mathbb{P}^e} \left[ Y^{\mathrm{do}(T=1)} \right],$$

since $Z_{\mathrm{Pa}(Y)}$ is a valid adjustment set. Then, we simplify the second treatment term in our estimand using the tower property the invariance of the conditional expectation:

$$\mathbb{E}_{\mathbb{P}^e} \left[ \frac{(Y - \bar{\mu}_1(X_{S_{\cancel{Y}}}))T}{\bar{\pi}(X_{S_{\cancel{Y}}})} \right] = \mathbb{E}_{\mathbb{P}^e} \left[ \frac{1}{\bar{\pi}(X_{S_{\cancel{Y}}})} \mathbb{E}_{\mathbb{P}^e} \left[ (Y - \bar{\mu}_1(X_{S_{\cancel{Y}}})) \mid X_{S_{\cancel{Y}}}, T = 1 \right] \mathbb{P}^e(T = 1 \mid X_{S_{\cancel{Y}}}) \right].$$

The RHS is equal to zero since $\mathbb{E}_{\mathbb{P}^e} \left[ Y \mid X_{S_{\cancel{Y}}}, T = 1 \right] = \bar{\mu}_1(X_{S_{\cancel{Y}}})$, $\mathbb{P}^e - \text{a.s.}$, as the invariance from Equation (10) holds.

With an analogous argument for the control term it follows that

$$\mathbb{E}_{\mathbb{P}^e} \left[ \frac{(Y - \bar{\mu}_0(X_{S_{\cancel{Y}}}))(1 - T)}{1 - \bar{\pi}(X_{S_{\cancel{Y}}})} \right] = 0,$$

and thus, the RAMEN estimator is equivalent to the ATE:

$$\theta^e(S_{\cancel{Y}}) = \mathbb{E}_{\mathbb{P}^e} \left[ Y^{\mathrm{do}(T=1)} \right] - \mathbb{E}_{\mathbb{P}^e} \left[ Y^{\mathrm{do}(T=0)} \right].$$

### A.3 IMPLEMENTATION DETAILS

In this section, we describe all the implementation details for our methodology.

**Estimation of the loss function** We have several choices when it comes to estimating our loss function, as there is a trade-off between statistical and computational efficiency. For instance, one can choose the linear time estimator proposed in Gretton et al. (2012, Section 6) or the efficient estimator proposed in Kim & Ramdas (2024) that runs in quadratic time. In this paper, we estimate

$$\mathbb{H}_e^2(S) := \mathbb{E}_{\mathbb{P}^e} \left[ \delta_S(Z; V) k \left( Z_S, \tilde{Z}_S^{\mathrm{os}} \right) \delta_S(\tilde{V}, \tilde{Z}^{\mathrm{os}}) \right]$$

using the cross U-statistic from Kim & Ramdas (2024), defined as

$$\hat{\mathbb{H}}_e^2(S) := \frac{2}{n} \sum_{i=1}^{n/2} h_S(Z_i, V_i),$$

$$\text{with } h_S(Z_i, V_i) := \frac{2}{n} \sum_{j=n/2+1}^{n} \delta_S(Z_i, V_i) k(Z_{i,S}, Z_{j,S}) \delta_S(Z_j, V_j).$$

Moreover, we would like the two loss functions, i.e., $J_Y$ and $J_T$, to be on the same scale to avoid any finite sample issues. Therefore, we standardize the cross U-statistic by dividing the empirical variance $\hat{\sigma} \left( \hat{\mathbb{H}}_e^2(S) \right)$, i.e. the finite sample estimate of the variance term

$$\sigma^2 \left( \hat{\mathbb{H}}_e^2(S) \right) := \mathbb{E}_{\mathbb{P}^e} \left[ \left( h_S(Z) - \mathbb{E}_{\mathbb{P}^e} [h_S(Z)] \right)^2 \right].$$

**Choice of kernel**  An important issue in practice is the selection of the kernel parameters. We used a Gaussian kernel in all of our experiments. We set the bandwidth of the kernel $\sigma$ to be the median distance between points $X$ in the pooled sample—this remains a heuristic similar to those described in Takeuchi et al. (2006), and the optimum kernel choice is an ongoing area of research.

### A.3.1  ALGORITHM 1: COMBINATORIAL SEARCH OVER SUBSETS

We now describe the concrete implementation of our first algorithm.

Since we know that $T$ is a parent of $Y$, we can simplify our loss function to incorporate this knowledge. Let us define the quantity $\delta_{y,t}(X_S, Y) := Y - \bar{\mu}_t(X_S)$, where $\bar{\mu}_t(X_S) := \mathbb{E}_{\bar{\mathbb{P}}}[Y \mid X_S, T = t]$. We can rewrite the Y-invariance loss function as follows

$$\min_{S \subseteq [d]} \max_{e \in \mathcal{E}, t \in \{0,1\}} \mathbb{E}_{\mathbb{P}^e} \left[ \delta_{y,t}(Y, X_S) k \left( X_S, X_S' \right) \delta_{y,t}(Y', X_S') \mid T = t \right].$$

Similarly, we define $\bar{\pi}(X_S) := \mathbb{E}_{\bar{\mathbb{P}}}[T \mid X_S]$ and minimize the T-invariance loss function

$$\min_{S \subseteq [d]} \max_{e \in \mathcal{E}} \mathbb{E}_{\mathbb{P}^e} \left[ \delta_t(T, X_S) k \left( X_S, X_S' \right) \delta_t(T', X_S') \right].$$

where $\delta_t(X_S, T) := T - \bar{\pi}(X_S)$.

We explain how to compute the adjustment set explicitly in Algorithm 1, assuming oracle access to the nuisance functions. In practice, nuisance functions can be estimated using the pooled data from all environments.

### A.3.2  ALGORITHM 2: GUMBEL TRICK

To deal with the computational infeasibility of searching over all possible subsets of covariates, we propose a continuous relaxation of the optimization problem that can be efficiently solved using gradient descent. The method involves using Gumbel sampling to create differentiable binary masks for covariate selection, which allows optimization via gradient descent. We present the continuous relaxation in Algorithm 2 for obtaining the invariance loss with respect to the node $T$; the algorithm can be extended analogously to minimize the invariance loss for $Y_1$ and $Y_0$. In this algorithm, we replace the max operator over the environments with an average to obtain a smoother loss function.

---

**Algorithm 1** Combinatorial search over subsets ($\hat{\theta}_{\Cup}$)

---

1: **Input:** Data $\{(X_e, Y_e, T_e)\}_{i=1}^{n_e}$, Nuisance functions: $\bar{\pi}, \bar{\mu}_0, \bar{\mu}_1$
2: **for each subset** $S \subseteq [d]$ **do**
3:     **for each environment** $e \in \mathcal{E}$ **do**
4:         Compute T-invariance loss using dataset $D^e$:

$$\hat{J}_S(D^e; T) \leftarrow \frac{4}{n^2} \sum_{i=1}^{n/2} \sum_{j=n/2+1}^{n} (T_i - \bar{\pi}(X_i)) k(X_{i,S}, X_{j,S})(T_j - \bar{\pi}(X_j))$$

$$\hat{J}_S(D^e; T) \leftarrow \frac{\hat{J}_S(D^e; T)}{\widehat{\text{Var}}\left(\hat{J}_S(D^e; T)\right)}$$

5:         Compute Y-invariance loss using dataset $D^e$ restricted to samples with $T = 1$:

$$\hat{J}_S(D^e; Y_1) \leftarrow \frac{4}{n^2} \sum_{i=1}^{n/2} \sum_{j=n/2+1}^{n} (Y_i - \bar{\mu}_1(X_i)) k(X_{i,S}, X_{j,S})(Y_j - \bar{\mu}_1(X_j))$$

$$\hat{J}_S(D^e; Y_1) \leftarrow \frac{\hat{J}_S(D^e; Y_1)}{\widehat{\text{Var}}\left(\hat{J}_S(D^e; Y_1)\right)}$$

6:         Compute Y-invariance loss using dataset $D^e$ restricted to samples with $T = 0$:

$$\hat{J}_S(D^e; Y_0) \leftarrow \frac{4}{n^2} \sum_{i=1}^{n/2} \sum_{j=n/2+1}^{n} (Y_i - \bar{\mu}_0(X_i)) k(X_{i,S}, X_{j,S})(Y_j - \bar{\mu}_0(X_j))$$

$$\hat{J}_S(D^e; Y_0) \leftarrow \frac{\hat{J}_S(D^e; Y_0)}{\widehat{\text{Var}}\left(\hat{J}_S(D^e; Y_0)\right)}$$

7:     **end for**
8:     Compute the worst environment losses:

$$\hat{J}_S(T) \leftarrow \max_{e \in \mathcal{E}} J_S(D^e; T), \;\; \hat{J}_S(Y_1) \leftarrow \max_{e \in \mathcal{E}} J_S(D^e; Y_1), \;\; \hat{J}_S(Y_0) \leftarrow \max_{e \in \mathcal{E}} J_S(D^e; Y_0)$$

9: **end for**
10: **Return:** $S_{\Cup} \leftarrow \text{argmin}_S \; \min\left(\hat{J}_S(T), \max(\hat{J}_S(Y_1), \hat{J}_S(Y_0))\right)$

---

---

**Algorithm 2** Gumbel trick for subset selection ($\hat{\theta}_{\text{insta}-\text{☙}}$)

---

1: **Input:** Data $\{(X_e, Y_e, T_e)\}_{e \in \mathcal{E}}$, temperatures $\tau_{\text{init}}, \tau_{\text{final}}$, interval $k$, rate $\alpha < 1$, learning rates $\eta_{\text{gate}}, \eta_{\text{nn}}$, epochs $n_{\text{epochs}}$
2: Initialize gate weights $w_\pi, w_y$ and neural networks $\theta_\pi, \theta_{y_1}, \theta_{y_0}$; Set $\tau \leftarrow \tau_{\text{init}}$
3: **for** epoch $= 1$ to $n_{\text{epochs}}$ **do**
4:     **if** epoch mod $k = 0$ **then**
5:         Update temperature: $\tau \leftarrow \max(\tau_{\text{final}}, \tau \cdot \alpha)$
6:     **end if**
7:     **for each environment** $e \in \mathcal{E}$ **do**
8:         Compute masks:

$$B_j^\pi = \text{sigmoid}((w_{\pi,j} + G_{1,j} - G_{2,j})/\tau) \quad \text{with } G_{1,j}, G_{2,j} \sim \text{Gumbel}(0, 1)$$
$$B_j^y = \text{sigmoid}((w_{y,j} + G_{1,j} - G_{2,j})/\tau) \quad \text{with } G_{1,j}, G_{2,j} \sim \text{Gumbel}(0, 1)$$

9:         Compute invariance losses:

$$\hat{J}_{w_\pi}(D^e; T) \leftarrow \frac{4}{n^2} \sum_{i=1}^{n/2} \sum_{j=n/2+1}^{n} (T_i - f_{\theta_\pi}(X_{i,B^\pi})) k(X_{i,B^\pi}, X_{j,B^\pi})(T_j - f_{\theta_\pi}(X_{j,B^\pi}))$$

$$\hat{J}_{w_\pi}(D^e; T) \leftarrow \frac{\hat{J}_{w_\pi}(D^e; T)}{\widehat{\text{Var}}\left(\hat{J}_{w_\pi}(D^e; T)\right)}$$

$$\hat{J}_{w_y}(D^e; Y_1) \leftarrow \frac{4}{n^2} \sum_{i=1}^{n/2} \sum_{j=n/2+1}^{n} (Y_i - f_{\theta_{y_1}}(X_{i,B^y})) k(X_{i,B^y}, X_{j,B^y})(Y_j - f_{\theta_{y_1}}(X_{j,B^y})) \quad \text{(using } D^e \text{ with } T = 1)$$

$$\hat{J}_{w_y}(D^e; Y_1) \leftarrow \frac{\hat{J}_{w_y}(D^e; Y_1)}{\widehat{\text{Var}}\left(\hat{J}_{w_y}(D^e; Y_1)\right)}$$

$$\hat{J}_{w_y}(D^e; Y_0) \leftarrow \frac{4}{n^2} \sum_{i=1}^{n/2} \sum_{j=n/2+1}^{n} (Y_i - f_{\theta_{y_0}}(X_{i,B^y})) k(X_{i,B^y}, X_{j,B^y})(Y_j - f_{\theta_{y_0}}(X_{j,B^y})) \quad \text{(using } D^e \text{ with } T = 0)$$

$$\hat{J}_{w_y}(D^e; Y_0) \leftarrow \frac{\hat{J}_{w_y}(D^e; Y_0)}{\widehat{\text{Var}}\left(\hat{J}_{w_y}(D^e; Y_0)\right)}$$

10:     **end for**
11:     Compute losses:

$$\hat{J}_{w_\pi} \leftarrow \frac{1}{|\mathcal{E}|} \sum_e \hat{J}_{w_\pi}(D^e; T), \quad \hat{J}_{w_y} \leftarrow \frac{1}{2|\mathcal{E}|} \sum_e \hat{J}_{w_y}(D^e; Y_1) + \hat{J}_{w_y}(D^e; Y_0)$$

12:     Update parameters:

$$w_\pi \leftarrow w_\pi - \eta_{\text{gate}} \nabla_{w_\pi} \hat{J}_{w_\pi}, \; \theta_\pi \leftarrow \theta_\pi - \eta_{\text{nn}} \nabla_{\theta_\pi} \hat{J}_{w_\pi}$$
$$w_y \leftarrow w_y - \eta_{\text{gate}} \nabla_{w_y} \hat{J}_{w_y}, \; \theta_{y_1} \leftarrow \theta_{y_1} - \eta_{\text{nn}} \nabla_{\theta_{y_1}} \hat{J}_{w_y}, \; \theta_{y_0} \leftarrow \theta_{y_0} - \eta_{\text{nn}} \nabla_{\theta_{y_0}} \hat{J}_{w_y}$$

13: **end for**
14: Determine subsets: $S_T \leftarrow \{i : w_{\pi,i} > 0\}, S_Y \leftarrow \{i : w_{y,i} > 0\}$
15: Calculate exact losses (as defined in Algorithm 1) for each subset $S \in \{S_T, S_Y\}$:

$$J_S(T) \leftarrow \frac{1}{|\mathcal{E}|} \sum_e \hat{J}_S(D^e; T), \; J_S(Y_1) \leftarrow \frac{1}{|\mathcal{E}|} \sum_e \hat{J}_S(D^e; Y_1), \; J_S(Y_0) \leftarrow \frac{1}{|\mathcal{E}|} \sum_e \hat{J}_S(D^e; Y_0)$$

16: **Return:** $S_{\text{insta}-\text{☙}} \leftarrow \text{argmin}_{S \in \{S_T, S_Y\}} \min(J_S(T), \max(J_S(Y_1), J_S(Y_0)))$

---

## B    EXTENDED RELATED WORK

We discuss here the different challenges associated with the problem of identifying and estimating treatment effects. Our focus is to highlight differences and similarities with our methodology—we leave out the orthogonal problem of statistical efficiency for the sake of clarity, and we refer the reader to Guo et al. (2022); Cheng et al. (2024) for a complete survey of methods.

**Covariate selection with pre-treatment covariates**    Several works have relaxed the causal suffi­ciency assumption, allowing for unobserved variables—as long as they are not confounders—while constraining all observed covariates to be pre-treatment. In this setting, the main challenge is M-bias (Sjölander, 2009), which makes adjusting for the full set of covariates not a viable solution. For instance, EHS (Entner et al., 2013) was one of the first methods to obtain partial identification of treatment effects in this setting, however, at the cost of computational inefficiency. Gultchin et al. (2020) propose a more efficient relaxation for EHS to circumvent the computational inefficiency. Further, several more recent works leverage anchor variables to obtain point identification in a com­putationally efficient way (Cheng et al., 2020; 2022b; Shah et al., 2022). In contrast, our setting is different since we do not assume that all observed covariates are pre-treatment.

**Covariate selection under causal sufficiency**    When all the variables in the causal graph are observed, the only challenge towards identifiability is the presence of post-treatment covariates that can introduce collider bias. Several methods have been proposed to tackle this setting—e.g. IDA (Maathuis et al., 2009) and its variants (Perković et al., 2017; Fang & He, 2020) aim to learn a complete graph from data and then infer a valid adjustment set from it to achieve identifiability. However, they suffer from computational inefficiency since they must first learn the entire causal graph, and they only achieve partial identification. More recently, Shi et al. (2021) consider the setting where multiple environments are available and apply invariant risk minimization (IRM) (Ar­jovsky et al., 2019) for treatment effect estimation. However, it is widely known that IRM re­quires many environments—linear in the number of covariates—to generalize well even in the linear regime (Rosenfeld et al., 2021). Finally, Wang et al. (2023) recently proposed a reinforcement learn­ing approach to identify the treatmente effect. In contrast, our approach achieves point identification while being computationally efficient and not requiring causal sufficiency.

**Identifiability in linear Gaussian SCMs**    In linear Gaussian structural causal models (SCMs), the structure of the causal graph imposes algebraic relationships among the entries in the covariance matrix of the associated distribution. Many researchers have exploited these relationships to de­rive graphical criteria for the identifiability of causal effects, even when some confounders remain unobserved. Specifically, several graphical criteria have been identified for deciding whether, in a given causal graph, a specific causal effect can be identified from the covariance matrix for almost all linear Gaussian SCMs compatible with the graph (Drton et al., 2011; Foygel et al., 2012; Weihs et al., 2018; Barber et al., 2022). In contrast, our approach does not assume linearity or Gaussianity, and instead, we leverage access to multiple heterogeneous data sources for identification.

**Combining data from multiple environments**    Given the challenges associated with estimating treatment effects using non-randomized data, several works propose detecting bias in the treatment effect estimated from observational data by leveraging randomized trials (Yang et al., 2023; Morucci et al., 2023; Hussain et al., 2022; 2023; Demirel et al., 2024; De Bartolomeis et al., 2024b;a), or mul­tiple observational studies (Karlsson & Krijthe, 2023; Mameche et al., 2024; Karlsson & Krijthe, 2025). In contrast, we leverage the heterogeneity across multiple data sources to identify and esti­mate treatment effects in settings without unobserved confounders.

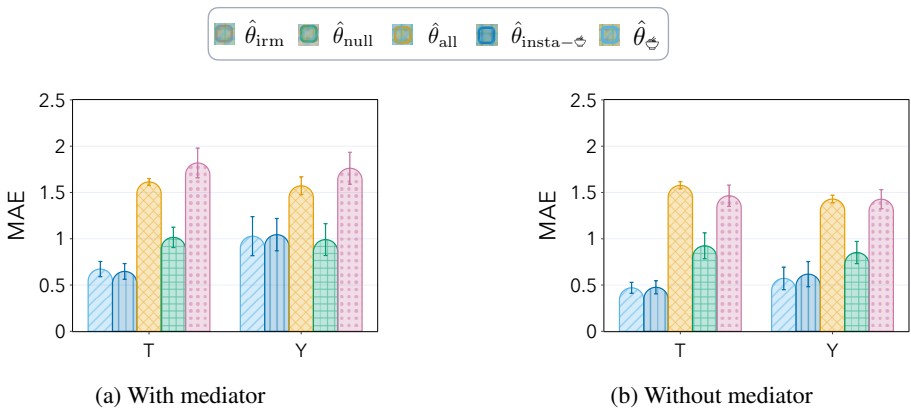

(a) With mediator           (b) Without mediator

Figure 6: Mean absolute error averaged across environments for the IHDP dataset with a descendant of the outcome $Y$ when different invariances are preserved ($T$ or $Y$). We consider the setting with (a) a mediator between $T$ and $Y$ and (b) without a mediator. We consider five environments with $n = 748$ points each; mean and standard error are reported over 20 runs.

## C   ADDITIONAL EXPERIMENTS

### C.1   ROBUSTNESS TO MEDIATORS

We study here the robustness of our method to violations of Assumption 3.1. More concretely, we show how the inclusion of a mediator between $T$ and $Y$ affects the ATE estimate for our method and the baselines in several settings. We consider the semi-synthetic experiment setup from Section 5.3, using a 2-dimensional descendant of $Y$ as the post-treatment variable. In Figure 6, we present results for two settings: when the treatment or the outcome is invariant across environments (complete experimental details in Appendix D.2). All baselines show slightly worse performance when a mediator is included. When $T$ is invariant, our method remains competitive and outperforms the baselines, as the parents of $T$ still form a valid adjustment set despite the mediator. However, both $\hat{\theta}_{\circlearrowleft}$ and $\hat{\theta}_{\text{insta}-\circlearrowleft}$ experience a significant drop in performance in the $Y$-invariance setup, i.e. when $T$-invariance is violated. This is expected, as in this scenario, we recover the parents of $Y$, which unfortunately also includes the mediator. A closer inspection of the selected subsets reveals that they usually include the mediator, thus failing to estimate the full effect of $T$ on $Y$. Instead, our method recovers the natural direct effect of $T$ on $Y$ (Pearl, 2022).

### C.2   ROBUSTNESS TO LACK OF INVARIANCE

Next, we examine the robustness of our method to violations of the invariance in Assumption 3.2. Specifically, we consider again the semi-synthetic experiments of Section 5.3 in the scenario where neither $T$- nor $Y$-invariance holds and there are post-treatment variables (i.e. not the independent noise setting). We provide the results in Figure 7. The performance of our method significantly worsens in this setting, with performance close to the $\hat{\theta}_{\text{null}}$ baseline, as it often recovers the empty set when no invariant node is present. Nonetheless, our method still outperforms $\hat{\theta}_{\text{irm}}$ in all the settings considered.

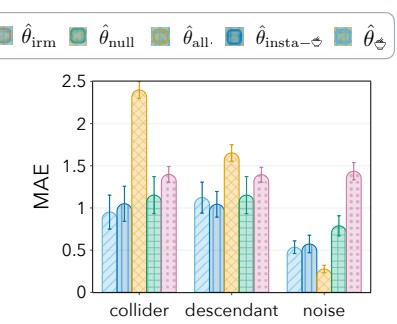

Figure 7: Mean absolute error averaged across environments for the IHDP dataset when no invariance is preserved. We consider five environments with $n = 748$ points each; mean and standard error are reported over 20 runs.

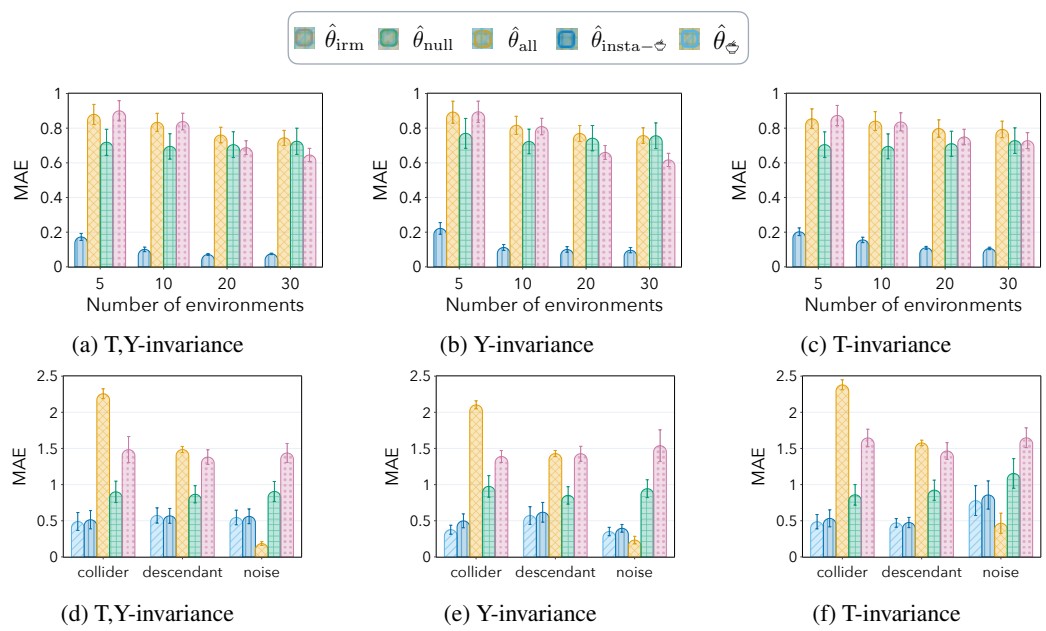

Figure 8: (**Row 1**) We plot the mean absolute error averaged across environments when: (a) no unobserved variables and both invariance w.r.t $T$ and $Y$ are preserved; (b) the parents of $T$ are unobserved but the invariance w.r.t. $Y$ is preserved; (c) the parents of $Y$ are unobserved but the invariance w.r.t $T$ is preserved. For all plots, we sample $n = 2000$ points for each environment; we report mean and standard error over 100 runs. (**Row 2**) Complete experimental results for the semi-synthetic setup described in Section 5.3 using the IHDP dataset. The plots show the mean absolute error averaged across environments for: (a) both $T$- and $Y$-invariance, (b) $Y$-invariance only, (c) $T$-invariance only. We consider five environments with $n = 748$ points each; mean and standard error are reported over 20 runs.

## C.3 ADDITIONAL RANDOM GRAPHS EXPERIMENTS

In Figure 8 (Row 1), we report the MAE averaged across environments for all the invariance settings. First, we observe that across all settings and numbers of available environments, our method significantly outperforms existing baselines. Most notably, $\hat{\theta}_{\text{insta}-\circleftarrow}$ achieves relatively small errors even with a limited number of environments. In contrast, $\hat{\theta}_{\text{irm}}$ requires a much larger number of environments to outperform the trivial baselines $\hat{\theta}_{\text{null}}$ and $\hat{\theta}_{\text{all}}$. Further, when the parents of $Y$ are unobserved, $\hat{\theta}_{\text{irm}}$ fails to surpass all trivial baselines, even with many environments—this outcome is expected, as the $Y$-invariance is broken in this case and $\hat{\theta}_{\text{irm}}$ lacks the double robustness.

## C.4 ADDITIONAL SEMI-SYNTHETIC EXPERIMENTS

We present the complete experimental results using the IHDP dataset (see Section 5.3) in Figure 8 (Row 2). Specifically, we evaluate our proposed method and the baselines under three conditions, where the two-dimensional variable $X_c$ acts as a collider (as described in the main text), descendant, or independent noise. For $T$-, $Y$-, and $T, Y$- invariance, the results align with those obtained in previous sections for linear synthetic experiments. Both $\hat{\theta}_\circleftarrow$ and its differentiable approximation, $\hat{\theta}_{\text{insta}-\circleftarrow}$, outperform the baselines in most settings. The sole exception is when the post-treatment variables are independent noise, where $\hat{\theta}_{\text{all}}$ achieves the best performance. In the case of $T$-invariance, both our method and $\hat{\theta}_{\text{irm}}$ exhibit slightly worse performance. $\hat{\theta}_{\text{irm}}$ generally underperforms, showing the highest error even under the independent noise setting. The $\hat{\theta}_{\text{null}}$ baseline demonstrates competitive performance overall, likely due to the relatively low influence of confounders in this setup.

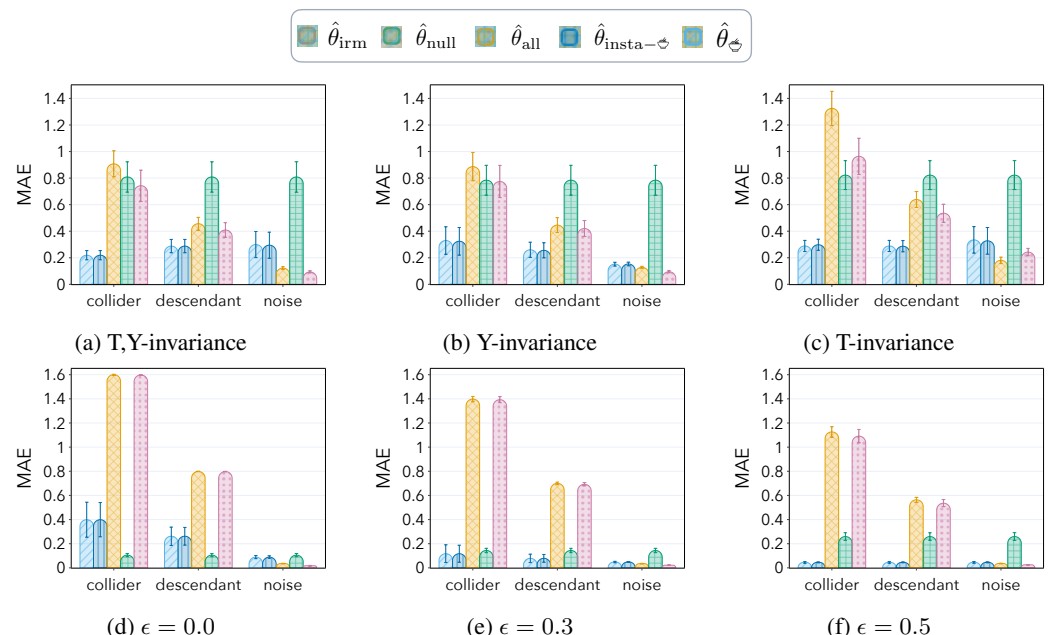

Figure 9: (**Row 1**) For all the plots: $n = 250$, $d = 3$, $|\mathcal{E}| = 5$. We plot the mean absolute error averaged across environments when: (a) both invariances are preserved; (b) the invariance w.r.t $Y$ is preserved; (c) the invariance w.r.t $T$ is preserved. We report mean and standard error over 20 runs. (**Row 2**) For all the plots: $n = 2500$, $d = 3$, $|\mathcal{E}| = 5$, and only the invariant w.r.t $Y$ is preserved. We plot the mean absolute error averaged across environments for different levels of heterogeneity in the data (higher $\epsilon$ corresponds to more heterogeneous data).

## C.5 ROBUSTNESS TO SMALL SAMPLE SIZE

In Figure 9 (a–c), we report the MAE averaged across environments for the three graphical models introduced in Figure 2 (Row 2). We can observe that our method remains competitive even in the small sample size regime: our method consistently outperforms all baselines when a collider or descendant is present. However, its performance declines in the edge case where the post-treatment variable is independent noise.

## C.6 ROBUSTNESS TO VIOLATIONS OF ASSUMPTION 4.1

We evaluate here the robustness of our method against violations of our identification condition (i.e. Assumption 4.1). To do so, we slightly modify the synthetic experiments presented in Figure 2: We introduce a parameter $\epsilon^2$ to control environment heterogeneity. If $\epsilon^2 = 0$, $X$ has the same distribution across environments. Therefore, there is no heterogeneity across environments, and Assumption 4.1 is violated. On the other hand, if $\epsilon^2 > 0$, the mean and variance of $X$ will shift across environments, with larger shifts as the parameter $\epsilon^2$ increases. Therefore, the heterogeneity across environments increases with $\epsilon^2$, and Assumption 4.1 is more likely to be satisfied.

**Example C.1** (Post-treatment variables). *Let $\mathcal{E}$ be the collection of environment indices. For each environment $e \in \mathcal{E}$, we first sample $U \sim \mathcal{N}(0, \epsilon^2 I_{d+1})$. Then, the data is given by*

$$X_{p,i} \sim \mathcal{N}(U_i, 0.5 + U_i^2), \ \text{for} \ i = 1, \ldots, d-1;$$
$$T \sim \text{Ber}\left(\sigma\left(\beta_t^\top X_p + \epsilon_t\right)\right), \ \text{with} \ \beta_t \sim \mathcal{N}(0, I_{d-1}) \ \text{and} \ \epsilon_t \sim \mathcal{N}(U_d, 0.5 + U_d^2);$$
$$Y = T + \beta_y^\top X_p + \epsilon_y, \ \text{with} \ \beta_y \sim \mathcal{N}(0, I_{d-1}) \ \text{and} \ \epsilon_y \sim \mathcal{N}(0, 1);$$
$$X_c = a \cdot T + b \cdot Y + \epsilon_c, \ \text{with} \ \epsilon_c \sim \mathcal{N}(U_{d+1}, 0.5 + U_{d+1}^2).$$

In Figure 9 (d–f), we plot the MAE for different levels of heterogeneity in the data. We can observe that both $\hat{\theta}_{\clubsuit}$ and $\hat{\theta}_{\text{irm}}$ suffer significantly when there is no heterogeneity ($\epsilon = 0.0$). Nevertheless, our method $\hat{\theta}_{\clubsuit}$ consistently outperforms $\hat{\theta}_{\text{irm}}$, even under strong violations of the identification condition. Moreover, $\hat{\theta}_{\clubsuit}$ remains competitive against all baselines when Assumption 4.1 is only weakly satisfied ($\epsilon = 0.3$).

## D EXPERIMENTAL DETAILS

Given an adjustment set, we estimate the ATE for each environment $e \in \mathcal{E}$ as follows

$$\hat{\theta}_S^e = \frac{1}{n} \sum_{(X_i, T_i, Y_i) \in D^e} \hat{\mu}_S^e(X_i, 1) - \hat{\mu}_S^e(X_i, 0) + \frac{(Y_i - \hat{\mu}_S^e(X_i, 1))T_i}{\hat{\pi}_S^e(X_i)} - \frac{(Y_i - \hat{\mu}_S^e(X_i, 0))(1 - T_i)}{1 - \hat{\pi}_S^e(X_i)},$$

where $\hat{\mu}_S^e(x, t) = \hat{\mathbb{E}}_{\mathbb{P}^e}[Y \mid T = t, X_S = x]$ and $\pi_S^e(x) = \hat{\mathbb{E}}_{\mathbb{P}^e}[T \mid X_S = x]$.

For $\hat{\theta}_{\mathrm{irm}}$, since the algorithm only learns the outcome function, we estimate the ATE as

$$\hat{\theta}_{\mathrm{irm}} = \frac{1}{n} \sum_{(X_i, T_i, Y_i) \in D^e} \hat{\mu}_S^e(X_i, 1) - \hat{\mu}_S^e(X_i, 0).$$

### D.1 SYNTHETIC EXPERIMENTS

We now describe the data generating process for our synthetic experiments in Section 5.1.

**Example D.1** (Post-treatment variables). *Let $\mathcal{E}$ be the collection of environment indices. For each environment $e \in \mathcal{E}$, we first sample $U \sim \mathcal{N}(0, I_{d+1})$. For $a, b \in \{0, 1\}$, we then observe the following variables:*

$$X_{p,i} \sim \mathcal{N}(U_i, U_i^2), \ \ for \ \ i = 1, \dots, d-1;$$

$$T \sim \mathrm{Ber}\left(\sigma\left(\beta_t^\top X_p + \epsilon_t\right)\right), \ \ with \ \ \beta_t \sim \mathcal{N}(0, I_{d-1}) \ \ and \ \ \epsilon_t \sim \begin{cases} \mathcal{N}(0, 1) & if \ T \ is \ invariant \\ \mathcal{N}(U_d, U_d^2) & else \end{cases};$$

$$Y = T + \beta_y^\top X_p + \epsilon_y, \ \ with \ \ \beta_y \sim \mathcal{N}(0, I_{d-1}) \ \ and \ \ \epsilon_y \sim \begin{cases} \mathcal{N}(0, 1) & if \ Y \ is \ invariant \\ \mathcal{N}(U_d, U_d^2) & else \end{cases};$$

$$X_c = a \cdot T + b \cdot Y + \epsilon_c, \ \ with \ \ \epsilon_c \sim \mathcal{N}(U_{d+1}, U_{d+1}^2).$$

Further, observe that for each choice of invariance, the post-treatment variable $X_c$ can either be a descendant of $Y$ ($a = 0$ and $b = 1$), a collider between $T$ and $Y$ ($a = 1$ and $b = 1$), or independent noise ($a = 0, b = 0$). Finally, under this data-generating process, the average treatment effect is constant across the environments, and it is given by $\theta^e = 1$, for all $e \in \mathcal{E}$.

**Random graph data generating process (Section 5.2)** We randomly draw a graph from the Erdös-Rényi random graph model with a density equal to $0.5$ and consider graphs with a total number of observed nodes $p = 20$. We do rejection sampling to exclude graphs that either contain mediators (as they violate Assumption 3.1) or do not contain at least a confounder. We then sample data from the resulting DAG via a linear structural causal model with Gaussian weights using the `causaldag` python library, with the only exception being the treatment variable $T$, which is generated by additionally applying a sigmoid function and then sampling from a Bernoulli distribution. We further post-process the graph, adding a post-treatment variable $X_c = Y + T$ and removing at random some parents of $T$ or $Y$ depending on which invariance we want to preserve. Therefore, we consider a challenging scenario with both a collider and unobserved variables. To sample data from multiple environments $e \in \mathcal{E}$, within each environment $e$, we apply a random uniform mean and variance shift to all the nodes in the graph, except for $T$ and $Y$.

**Implementation details** We implement our method, $\hat{\theta}_{\mathrm{insta} - \circleddash}$, by performing a hyperparameter search over the following parameters at each iteration: learning rate in the range $[0.001, 0.01, 0.1]$, initial temperature values of $[0.5, 0.8, 1.0]$, and annealing rates of $[0.9, 0.95, 0.99]$. The optimal combination of these hyperparameters is selected based on minimizing both T-invariance and Y-invariance loss. The outcome functions for $\hat{\theta}_{\mathrm{all}}$, $\hat{\theta}_{\circleddash}$, $\hat{\theta}_{\mathrm{insta} - \circleddash}$ and $\hat{\theta}_{\mathrm{irm}}$ are estimated using a linear regression model. Logistic regression is used for propensity score estimation.

### D.2 INFANT HEALTH AND DEVELOPMENT PROGRAM (IHDP) DATASET

The Infant Health and Development Program (IHDP) dataset is a randomized controlled trial focusing on low-birth-weight, premature infants. For our analysis, we keep six continuous covariates from Dorie (2016), representing the child's birth weight, head circumference at birth, number of weeks pre-term, birth order, neonatal health index, and mother's age at birth.

Instead of adopting the treatment and outcome functions from Dorie (2016), we simulate a more challenging scenario inspired by Kang & Schafer (2007). In this setting, each covariate assigned to the treatment ($T$) or outcome ($Y$) undergoes a transformation using a predefined set of complex functions similar to those encountered in real-world applications. We introduce the following relationships:

- Confounders: Three of the six covariates are randomly selected to act as confounders, affecting both $T$ and $Y$.

- Other pre-treatment covariates: The remaining covariates are assigned to affect either $T$ or $Y$, but not both.

- Post-treatment covariates: We include a two-dimensional post-treatment covariate, denoted as $Z$, whose generation is detailed below.

- Environmental variation: To introduce variation across environments, we (i) randomly make a parent of either $T$ or $Y$ unobserved (the same one across environments) and (ii) introduce environment-specific shifts, as detailed below. We apply both to the same node ($T$ or $Y$) so that the other remains invariant.

- We set ATE $= 2$ for all environments.

**Modeling of $T$ and $Y$**   For each covariate $X_i$ affecting $T$, we apply a randomly chosen transformation $g_T^{(i)}(x)$ from the following set:

$$g_T^{(i)}(x) \in \left\{ 0.5 \log(|x| + 1), \ \left(\frac{x}{2}\right)^2, \ x + 0.2, \ \exp\left(\frac{x}{2}\right) \right\}.$$

We then compute the logits for the treatment assignment as:

$$T_{\text{logits}} = \sum_i \beta_T^{(i)} g_T^{(i)}(X_i),$$

where $\beta_T^{(i)}$ are coefficients sampled independently from a uniform distribution $\beta_T^{(i)} \sim \mathcal{U}(-0.5, 0.5)$. The binary treatment $T$ is obtained by applying a sigmoid function to $T_{\text{logits}}$ and sampling from a Bernoulli distribution:

$$P(T = 1) = \sigma(T_{\text{logits}}), \quad T \sim \text{Bernoulli}(P(T = 1)),$$

where $\sigma(x) = \frac{1}{1+e^{-x}}$ is the sigmoid function.

Similarly, for each covariate $X_j$ affecting $Y$, we apply a randomly chosen transformation $g_Y^{(j)}(x)$ from the set:

$$g_Y^{(j)}(x) \in \left\{ 2 \log(|x|), \ \left(\frac{x}{2}\right)^2, \ x + 1, \ \exp\left(\frac{x}{2}\right) \right\}$$

The outcome $Y$ is then computed as:

$$Y = \sum_j \beta_Y^{(j)} g_Y^{(j)}(X_j),$$

with coefficients $\beta_Y^{(j)}$ sampled from $\beta_Y^{(j)} \sim \mathcal{U}(-2, 2)$.

**Incorporating environment-specific shifts**    To introduce environment-specific variability, we define a hidden variable $U$ that modifies the pre-treatment and post-treatment covariates, outcome, and treatment assignment across different environments. The environments are indexed by $u = 0, 1, 2, 3, 4$. For each environment, we introduce shifts dependent on $u$.

We first sample coefficients:

$$\beta_{\text{inv}} \sim \mathcal{U}(0.5, 1.0), \quad \beta_X \sim \mathcal{U}(0.5, 1.0).$$

For each environment $u$, the shifts are generated as:

$$\Delta_{\text{inv}} = u \cdot \beta_{\text{inv}} + \epsilon_{\text{inv}}, \ \Delta_X = u \cdot \beta_X + \epsilon_X, \ \Delta_{\text{post}} = u \cdot \beta_X + \epsilon_{\text{post}},$$

where all $\epsilon_{inv}, \epsilon_X, \epsilon_{post}$ are independently sampled from $\mathcal{N}(0, 1)$.

Then, for each environment, the covariates are modified:

$$X = X^0 + \Delta_X,$$

where $X^0$ represents the original covariate values.

Either $Y$ or $T$ is also shifted, depending on the invariance we aim to preserve:

If invariance in $T$ :    $Y = Y^0 + \Delta_{\text{inv}},$    else if invariance in $Y$ :    $T_{\text{logits}} = T^0_{\text{logits}} + \Delta_{\text{inv}},$

while we add $\mathcal{N}(0, 1)$ to the invariant node.

**Generation of post-treatment variables $X_c$**    For each environment, we generate a two-dimensional post-treatment variable $X_c$ as follows:

- **Collider:**
$$X_c = Y + T + \epsilon_{\text{post}}, \quad \epsilon_{\text{post}} \sim \mathcal{N}(\Delta_{\text{post}}, I_2).$$

- **Descendant:**
$$X_c = Y + \epsilon_{\text{post}}, \quad \epsilon_{\text{post}} \sim \mathcal{N}(\Delta_{\text{post}}, I_2).$$

- **Independent Noise:**
$$X_c = \epsilon_{\text{post}}, \quad \epsilon_{\text{post}} \sim \mathcal{N}(\Delta_{\text{post}}, I_2).$$

**Inclusion of mediators**    In some settings, we introduce an additional mediator variable influenced by $T$:

$$\text{Mediator} = \beta_{\text{med}} \cdot T + \epsilon_{\text{med}}, \quad \beta_{\text{med}} \sim \mathcal{U}(-1.0, 1.0), \quad \epsilon_{\text{med}} \sim \mathcal{N}(0, 1).$$

The outcome $Y$ is then adjusted:

$$Y = Y + \text{Mediator}.$$

**Summary of data generation process**    For each environment:

1. Modify covariates: $X = X^0 + \Delta_X$.

2. Compute treatment: $T_{\text{logits}} = \sum_i \beta_T^{(i)}, g_T^{(i)}(X_i)$    $T \sim \text{Bernoulli}(\sigma(T_{\text{logits}}))$.

3. Compute outcome: $Y = \sum_j \beta_Y^{(j)}, g_Y^{(j)}(X_j)$.

4. Apply environmental shift to $Y$ or $T$ and hide a parent of $Y$ or $T$ (we hide the same parent for all environments).

5. Include the ATE $= 2.0$ in the outcome $Y$

6. If applicable, generate mediator and adjust $Y$.

7. Generate post-treatment variables $Z$.

**Implementation details**   We implement our method, $\hat{\theta}_{\text{insta}-\circlearrowleft}$, by performing a hyperparameter search over the following parameters at each iteration: learning rate in the range [0.001, 0.01, 0.1], initial temperature values of [0.5, 0.8, 1.0], and annealing rates of [0.9, 0.95, 0.99]. The optimal combination of these hyperparameters is selected based on the minimization of both T-invariance and Y-invariance loss. The outcome and treatment assignment functions for both $\hat{\theta}_{\text{all}}$ and $\hat{\theta}_{\text{insta}-\circlearrowleft}$ are estimated using XGBoost. For these models, we set the number of estimators to 1,000, the learning rate to 0.01, and the maximum tree depth to 6. For the non-linear IRM baseline, we employ the TARNet architecture (Shalit et al., 2017), which consists of a shared representation with a single hidden layer of 200 neurons, followed by two hypothesis-specific hidden layers, each with 100 neurons. Logistic regression is used for propensity score estimation.

## D.3   CATTANEO2

The Cattaneo2 dataset (Cattaneo, 2010) studies the effect of maternal smoking on newborn birth weight. We consider 21 covariates, including maternal and paternal age and education, marital status, maternal foreign status, Hispanic origin, alcohol consumption, receipt of prenatal care and the number of prenatal visits, whether the mother had previous children who died, an indicator for low birth weight, months since last birth by the mother, birth month, indicator for whether the baby is first-born, and other variables for which full details are unavailable. The treatment is a binary indicator of smoking status, with 864 mothers in the treatment group and 3,778 in the control group. The outcome is a continuous variable representing birth weight, which we normalize to the interval $[0, 1]$. We exclude the month of birth from the observed features and instead use it to define the environments, creating four environments corresponding to the four quarters of the year.

**Implementation details**   We implement our method, $\hat{\theta}_{\text{insta}-\circlearrowleft}$, using the following hyperparameters: the number of epochs is set to 700, patience to 100, learning rate to 0.1, initial temperature to 1.0, and annealing rate to 0.9. This configuration was chosen because it provided robust and favorable results across experiments, specifically in minimizing T- and Y-invariance losses. All other hyperparameters are kept from previous experiments. The outcome and treatment assignment functions for both $\hat{\theta}_{\text{all}}$ and $\hat{\theta}_{\text{insta}-\circlearrowleft}$ are estimated using XGBoost, with the number of estimators set to 1,000, learning rate to 0.01, and maximum depth to 6. For the non-linear IRM implementation, we use the TARNet architecture, as in the IHDP experiments.

