# OpenReview forum: "Doubly robust identification of treatment effects from multiple environments"
_ICLR.cc/2025/Conference — ICLR 2025 Poster_

### Official Review · Reviewer_6ghW · 2024-10-22

**Soundness:** 2
**Presentation:** 3
**Contribution:** 2
**Rating:** 6
**Confidence:** 3

**Summary:**

This paper provides a new doubly robust identification framework given multiple data sources, in the sense that, it is able to identify the average treatment effect if in a causal DAG, the parent node of treatment or outcome is fully observed and conditional distribution of either treatment or outcome given their parents are the same across all data sources, without knowing which.

To identify the adjustment set, the paper proposed two losses based on the minimax problem outlined from the moment condition in the assumption above.

On the sample level, the paper conduct simulations to examine the performance of their proposed RAMEN estimator.

**Strengths:**

The presentation of the paper is clear. The problem tackled seems interesting. I am not entirely familiar with the literature on this direction so if I believe the contributions of other literature that this paper listed, I think the idea is novel in the literature.

**Weaknesses:**

1. I think one of key ingredient of this paper is Assumption 4.1. I found this assumption somewhat questionable and hard to believe.
 a. Can you provide what it means under some concrete real-data examples? Maybe explain for your real-data application specifically?
 b. Can you comment the testability and falsifiability of the assumption? Can one touch slightly or comment on some sort of sensitivitiy analysis you can imagine?
 c. Intuitively, not only different data sources should be heterogeneous, but the magnitude matter, especially in estimation when you identified the S_opt. So in the simulation, maybe you can add a sensitivity parameter to represent the strength of heterogeneity of data sources , and then twist that parameter (from zero (Assumption 4.1 fails) to strong) and see what happens?


2. Apart from 1c above examining assumption 4.1, I think there are multiple angles the simulation can be strengthened, so that readers can better judge the value and contribution of this work.
For example, to examine assumption 3.3, can you check a fourth setting where both when both (a) and (b) fails. This is a common practice when evaluation classical doubly robust estimators. We expect RAMEN will fail under this setting, but it can help me to justify the difficulty of your simulations setting. For example, if RAMEN even performs reasonably well under the 4th setting, it means the simulation is too easy and failure of (a) or (b) creates not enough difficulty. I think a reasonably setting would be to combine the scenario when one of Assumption 3.3 (a) and Assumption 3.3 (b) fails in your simulation setting (b) and (c) into a case that Assumption 3.3 fails.

3. In Section 5.4, I found using 4 trimesters of birth as different environment doubtful. Are they just repeated measure of the same pregant women for 4 times? Can you comment on what Assumption 3.3 and 4.1 means in your real-world experiment?
 a. Clarify if these are indeed repeated measures or separate groups of women.
 b. Explain how Assumptions 3.3 and 4.1 are expected to hold in this specific context.
 c. Suggest alternative ways to define environments in this dataset if trimesters are not appropriate.

**Questions:**

Needs clarification:
1. You assumed no presence of observed mediators (Assumption 3.2) but keeps emphasizing that the paper allows post-treatment variables and unmeasured variables, so do you mean you allow either unmeasured confounders, unmeasured mediators, or colliders (can be either observed or unobserved);
2. In Assumption 3.1, you said that \eps is an exogeneous noise vector following the joint distribution P_\eps^e over p independent variables. But on Page 4 line 202, you said "our setting does not require independence of the noise variable", is this a contradiction?
3. Page 2 line 83: "We then provide the first, to our knowledge, doubly robust identification guarantees for treatment effect in the presence of both post-treatment and unobserved variables." This contribution is misleading to readers. This approach is not the first approach to handle both post-treatment and unobserved variables, but rather the first doubly robust one (if I understood correctly). For example, for the "valid adjustment set" approach, as long as practitioners know this set, it also allows both post-treatment and unobserved variables in the DAG.

Address a limitation:
1. In abstract "Notably, RAMEN achieves doubly robust identification: we identify the treatment effect if either the causal parents of the treatment or those of the outcome are observed. " This needs more clarification because the doubly robust assumption not only requires either parent is observed but homogeneity of condiitonal distributions across sources of bias.
2. Solving a minimax problem can be difficult and slow. Can you add comments on the latency (speed) of running your estimator?
 a. Please provide specific runtime measurements for your method on the datasets used in the paper.
 b. Compare these runtimes to those of the baseline methods.
 c. Discuss how the runtime scales with dataset size and number of covariates.

---

> ### Author Response · Authors · 2024-11-16
>
> We thank the reviewer for their thoughtful feedback and we appreciate the acknowledgment of our work’s clarity and novelty.
>
> Below, we respond to the specific points raised.
>
> **[W1 Failure of Assumption 4.1]** We acknowledge that Assumption 4.1 is relatively strong. However, we emphasize that *it is implied by well-established assumptions in the invariance literature*, most notably the simultaneous noise intervention assumption introduced in [1] (Theorem 2, Assumption iii).
>
> Moreover, *the assumption can be falsified* in certain settings. For example, if practitioners know that an adjustment set  is invalid (e.g., it excludes a known confounder), they can test whether the conditional means of and shift across environments. If the conditional means are not shifting then the assumption is falsified.
>
> Nevertheless, we sincerely appreciate the reviewer’s suggestion (also raised by **Rev r3F3**) to examine the effects of violating this assumption. As suggested, we add a sensitivity parameter to represent the strength of heterogeneity of data sources, and then twist that parameter from zero (Assumption 4.1 fails) to high values (Assumption 4.1 is strongly satisfied). We have added these additional experiments in **Appendix C.6** (of the revised version).
>
> The main takeaway from these ablations is that both our method and previous methods relying on invariance (e.g., [2) perform poorly when the assumption is violated. However, our method outperforms IRM [2] even in this adversarial setting and is competitive with other baselines when the assumption is only weakly satisfied (i.e., small heterogeneity across environments).
>
>
> **[W2 Failure of Assumption 3.3]**  We agree with the reviewer that examining the impact of fully violating Assumption 3.3 is valuable. These results were already included in our paper (see **Appendix C.2**), where we show that, in cases of full violation, both our method and IRM fail due to the lack of invariance to exploit. The resulting error of both methods is significantly high, further suggesting that our simulation setting is not overly simplified.
>
> **[W3 Trimester of birth]** We appreciate the reviewer's comments and agree that using the trimester of birth as the environment variable may not be ideal (i.e. it may not satisfy our identification assumptions). However, we stress that *it is challenging to find publicly available real data with multiple environments* (e.g., data from many different hospitals). Geographic location or hospital indicator could serve as better environment variables that  introduce more significant shifts. Unfortunately, our data lack such information.
>
> Finally, we stress that *there are no repeated samples* in our experiment: each data point represents a unique individual that gave birth in a specific trimester of the year.
>
>
>
> **[C1 Post-treatment and unobserved variables]** That's a great question. Our setting accommodates *simultaneously* unobserved mediators, unobserved and observed colliders, and unobserved variables that are not confounders (as identifiability of the treatment effect would otherwise be fundamentally impossible). For instance, we allow for unobserved parents of the outcome, a scenario where previous methods like IRM [2] would fail.
>
> **[C2 Independence of noise]** We apologize for a slightly imprecise formulation. Indeed, the exogenous noise variables in Assumption 3.1 are assumed to be independent. However, the DAG in Assumption 3.1 is over both observed and unobserved variables. This DAG induces a corresponding "observed" DAG with noise variables that might be dependent. Thus, the setting is more general than [1] where the graph is assumed to be fully observed.
>
> Accordingly, we have added a footnote in line 215 to avoid any ambiguities.
>
>
> **[C3 Contribution]** Thank you for pointing this out. We implicitly assume in this sentence that a valid adjustment set is not known (which is often the case in practice)—if it were, identifiability would follow directly by definition. We believe this phrasing is correct and highlights the novelty of our method, which, to our knowledge, is the first to identify the treatment effect in the presence of both post-treatment and unobserved variables *when a valid adjustment set is not known*.
>
>
> [1] Peters, Jonas, Peter Bühlmann, and Nicolai Meinshausen. "Causal inference by using invariant prediction: identification and confidence intervals." Journal of the Royal Statistical Society Series B: Statistical Methodology 78.5 (2016): 947-1012.
>
> [2] Claudia Shi, Victor Veitch, and David Blei. Invariant representation learning for treatment effect estimation. Uncertainty in Artificial Intelligence, 2021

---

> > ### Author Response · Authors · 2024-11-16
> >
> > **[L1 Abstract clarification]** Thank you for the feedback. Unfortunately, at the abstract level, it is challenging to introduce all the required assumptions. Therefore, we focus on the non-standard assumptions in the abstract and introduce the remaining assumptions (that are well established in the literature) early in the main text.
> >
> > **[L2 Minimax problem]** Thanks for the great question. We would like to clarify that solving a hard minimax problem is not necessary for our method. The minimax problem in Equation 4, which can typically be difficult and slow to solve, is greatly simplified through the use of the kernel trick, as explained in Section 4.3.
> >
> > The main computational bottleneck comes from the kernel matrix computation, which has a computational complexity of $O(n^2 d)$, where $n$ is the number of samples and $d$ is the number of covariates. While this can be slow for very large datasets, the data sizes typically used in treatment effect estimation—usually around 1k to 20k samples—are not large enough to make this a limiting factor. Therefore, we expect our method to be computationally feasible in most practical settings.
> >
> > Regarding runtime, we do not provide specific measurements as our method can easily run on a MacBook Pro within a few minutes to a few hours, depending on the number of covariates and sample size. Providing specific runtime measurements would require us to re-run all our experiments. We understand the importance of computational complexity considerations, but since practitioners typically run the method once and prioritize estimation accuracy over small differences in runtime, we believe re-running these experiments may not provide significant insights.

---

> > > ### Comment · Reviewer_6ghW · 2024-11-25
> > >
> > > Thanks for the author's patient and thorough responses. These have mostly clarified my concerns. I have changed my score accordingly.

---

### Official Review · Reviewer_BoRS · 2024-11-01

**Soundness:** 3
**Presentation:** 2
**Contribution:** 3
**Rating:** 8
**Confidence:** 4

**Summary:**

The paper proposes RAMEN, a method that leverages multiple environments to achieve doubly robust identification of the ATE in the presence of post-treatment and unobserved variables. Empirical evaluations across synthetic, semi-synthetic, and real-world datasets show that the proposed method significantly outperforms existing methods.

**Strengths:**

1. This paper estimates causal effects in the presence of post-treatment and unobserved variables.
2. The paper introduces a novel double robustness property.
3. The authors demonstrate their method's effectiveness through extensive experiments on synthetic, semi-synthetic, and real-world datasets.

**Weaknesses:**

1. In the introduction, the explanation of valid and invalid adjustment sets lacks specific examples(such as in advertising recommendations or in the healthcare field), and it is difficult to understand the corresponding scenarios based only on the cause graph.
2. RAMEN should satisfy the positivity and ignoreability assumptions, which are not given in the problem setting of the paper.
3. There are many symbols and formulas in the paper. It may be better to list a symbol table.
4. The experimental evaluation metrics(such as PEHE[1] or ATE[1]) and comparison algorithms(such as[1]) are insufficient.

[1]Shalit U, Johansson F D, Sontag D. Estimating individual treatment effect: generalization bounds and algorithms[C].ICML’2017.

**Questions:**

1. What are the advantages of the proposed method compared with methods using neural networks, such as the method in the literature [1][2].
2. How is the number of samples in different environments determined in synthetic data experiments? To vary the number of samples per environment, it is recommended that sensitivity analysis experiments be added to synthetic datasets.

[1]Shalit U, Johansson F D, Sontag D. Estimating individual treatment effect: generalization bounds and algorithms[C].ICML’2017.

[2]Shi C, Blei D, Veitch V. Adapting neural networks for the estimation of treatment effects[C]. NIPS’2019.

---

> ### Author Response · Authors · 2024-11-13
>
> We are grateful to the reviewer for recognizing the strengths of our paper, including the importance of estimating treatment effects in the presence of post-treatment and unobserved variables, the introduction of a novel double robustness property, and our extensive experimental validation.
>
> We now address the raised weaknesses and questions.
>
> **[Q1 Advantages compared to (1) and (2)]** Our method differs fundamentally from [1] and [2] in its objectives. While these neural network approaches focus on estimation given a covariate set that satisfy ignorability, *our method addresses the prerequisite challenge of identifying an adjustment set that satisfies ignorability*. Importantly, our approach is *compatible with their estimation techniques* - once a valid adjustment set is identified, the neural networks from [1] and [2] can be used in the estimation stage.
>
>
> **[Q2 Sensitivity to sample size]** We agree that analyzing sensitivity to sample size is important. In the revised version, we have added additional synthetic experiments (**Appendix C.5, Figure 9**) focusing on the small sample size regime ($n=250$) since the synthetic experiments in our original submission used relatively large sample sizes.
>
> **[W2 Assumptions]** We want to emphasize that our method does not require *ignorability with respect to the full set of covariates*, as the covariate set might contain e.g. colliders. This allows our approach to be applied in settings where traditional methods might fail due to the presence of post-treatment variables (e.g. [1] and [2]). Further, we have explicitly stated the positivity assumption in Theorem 1 rather than in the problem setting to be transparent about the assumptions required to identify the ATE.
>
> **[W1 Examples]** We appreciate the suggestion to include concrete examples. While we focused on an abstract causal graph in the introduction for clarity, post-treatment variable bias appears frequently in real-world settings. The most notable example in healthcare is the birth-weight paradox: Studies found that among low birth-weight infants, those born to smokers had lower mortality risk than those born to non-smokers -- seemingly contradicting the overall relationship between maternal smoking and infant mortality. This paradox arose from inappropriately controlling for birth-weight, a post-treatment variable affected by maternal smoking.
>
> **[W4 Evaluation metrics]** The focus of our paper is on identifying the average treatment effect (ATE), which is the causal quantity of interest in most scientific inquiries. While extending the evaluation to the conditional average treatment effect (CATE) is an interesting future direction, we believe that our current scope is not limited. Identifying and estimating the ATE is already a challenging problem in settings where ignorability (with respect to the full set of covariates) is not satisfied.

---

> > ### Author Response · Authors · 2024-11-24
> >
> > Dear Reviewer BoRS,
> >
> > We hope our rebuttal has addressed your concerns and answered your questions. As the discussion period comes to a close, we would like to kindly ask if you have any further questions or concerns.
> >
> > Thank you once again for your time and thoughtful feedback!

---

> > > ### Comment · Reviewer_BoRS · 2024-11-25
> > > **Response to Authors**
> > >
> > > Thanks for the authors‘ reply. They have solved most of my concerns, and I am willing to improve the score.

---

### Official Review · Reviewer_oUQ4 · 2024-11-03

**Soundness:** 4
**Presentation:** 4
**Contribution:** 3
**Rating:** 8
**Confidence:** 3

**Summary:**

This paper considers a novel setting in which data is collected from heterogeneous environments, aiming to identify causal effects for each environment without prior knowledge of the causal graph. Under certain assumptions, the authors propose two algorithms to identify the target causal quantities. The effectiveness of this approach is demonstrated through extensive experiments.

**Strengths:**

1- The paper is well-written, and related work is thoroughly discussed. Additionally, the connection between the paper’s assumptions and previous work is clearly presented, for example, following Assumptions 3.3 and 4.1.

2- Various experiments have been conducted, demonstrating the significance of RAMEN.

**Weaknesses:**

1- The focus of the paper is solely on the identification of treatment effect; therefore, there is no analysis of sample complexity for the proposed algorithm.

**Questions:**

1- Could you discuss the point mentioned above?

2- What does “Descendant” mean in Figure 2?

3- Could you elaborate on Lines 264 and 278? They are not clear to me.

4- Regarding Theorem 1, we understand that the quantity is identifiable under certain assumptions. However, if some assumptions are not satisfied, can you demonstrate that the causal effect is not identifiable? This would be similar to the concept of completeness in the causal effect identification literature.

---

> ### Author Response · Authors · 2024-11-14
>
> We thank the reviewer for their thoughtful review and positive assessment of our paper. We appreciate their recognition of the clarity of our writing, the thorough discussion of related work, and the connections we drew between our assumptions and existing invariance assumptions in the literature.
>
> Below, we address the raised questions.
>
>
> **[Q1 Sample complexity]** Our focus in this paper was on identifiability, which is a challenging problem in itself. However, we strongly agree that sample complexity is a valuable direction for future work. We believe that the standard results from the double machine learning literature should apply to our estimator—if the nuisances (propensity score and outcome function) are estimated at the classic $o_{\mathbb P}(n^{-1/4})$ rate,  our estimator should be asymptotically normal, allowing for valid (asymptotic) confidence intervals.
>
>
> **[Q2 Descendant in Figure 2]** In Figure 2, "Descendant" refers to the underlying causal graph where the node $X_c$  is a descendant of the outcome $Y$ but not of the treatment $T$  (to ensure that it is not a collider).
>
> **[Q3 Line 264]** In line 264, we raise a crucial point regarding the behavior of our method when distributions are not shifting across environments. Specifically, if $\mathbb P^e = \mathbb P^f$ for all $e,f \in \mathcal E$, any adjustment set would minimize our objective in Equation 4. This occurs because $E_{\mathbb P^e}[V | Z_S] =  E_{\mathbb P^f}[V | Z_S]$ holds true for any environments $e,f \in \mathcal E$ and any subset of covariates $S$.
>
> Essentially, when the distributions are the same across environments, we cannot use the invariance principle to differentiate between valid and invalid adjustment sets (because everything is invariant).
>
> **[Q3 Line 278]** In line 278, we discuss the number of environments required to satisfy the heterogeneity conditions. We note that if environments are generated through single-node interventions (i.e. each environment is sampled from the same distribution with an intervention on a different node), we would need a number of environments in the order of the number of nodes in the causal graph.
>
> **[Q4 Completeness]** That’s a great question. We considered this and briefly discussed it in Appendix A.1. Unfortunately, our assumption is not minimal: in some cases, it might still be possible to find a valid adjustment set using the observed parents of either $T$  or $Y$ (or both), even if the full set of parents is not observed (see the causal graph in Figure 5, for example). However, this set cannot be recovered using invariance approaches, as neither $ T $ nor $ Y $ are invariant across environments when some of their parents are unobserved and their distribution shifts.

---

> > ### Comment · Reviewer_oUQ4 · 2024-11-25
> >
> > Thanks for your response and clarification. I’ll maintain my current score.

---

### Official Review · Reviewer_r3F3 · 2024-11-04

**Soundness:** 3
**Presentation:** 3
**Contribution:** 3
**Rating:** 6
**Confidence:** 3

**Summary:**

This work addresses the bias arising from adjusting for bad controls in observational causal inference by leveraging invariance conditional properties of either the treatment or the outcome across multiple environments. The methodology includes two practical solutions and they are validated across synthetic, semi-synthetic, and real-world datasets.

**Strengths:**

1. The issue of bad controls is important in observational causal inference. The proposed approach of excluding them using multi-environment data appears to be a novel idea.
3. Comprehensive simulations are done to demonstrate the performance and robustness of the proposed algorithms.
4. The paper is well-written and clear.

**Weaknesses:**

1. The identification assumptions seem strong and the real-world applicability might be constrained. Are any parts of the assumptions testable using observed data, or can any robustness checks or sensitivity analyses be performed? Have you tested how violations of Assumption 4.1 impact the results?
2. To provide more convincing results regarding the method's usefulness in real-world applications, could you elaborate more on the selected controls by the algorithm in the birthweight dataset? Additionally, why is it difficult to exclude those potential bad controls or colliders based solely on domain knowledge?
3. Assumption 4.1 can be renamed since it's one of the identification assumptions.

**Questions:**

1. In figure 2a and 2c, if $X_c$ is only a descendant of $Y$, why does adjusting for both covariates lead to bias?
2. How are the standard errors calculated and why are they significantly higher for the proposed algorithm compared to the baselines in the application?
3. Can this approach be generalized to non-binary treatments and other estimands?

---

> ### Author Response · Authors · 2024-11-16
>
> We thank the reviewer for their thoughtful feedback and recognition of our contributions. We appreciate the acknowledgment of our work’s clarity, scalability, and the importance of addressing bad controls in causal inference.
>
> Below, we respond to the specific points raised.
>
> **[W1 Assumption 4.1]** We acknowledge that Assumption 4.1 is relatively strong. However, we emphasize that *it is implied by well-established assumptions in the invariance literature*, most notably the simultaneous noise intervention assumption introduced in [1] (Theorem 2, Assumption iii).
>
> Moreover, *the assumption can be falsified in certain settings*. For example, if practitioners know that an adjustment set $Z$ is invalid (e.g., it excludes a known confounder), they can test whether the conditional means of $Y|Z$ and $T|Z$ shift across environments. If the conditional means are not shifting then the assumption is falsified.
>
>
> Nevertheless, we sincerely appreciate the reviewer’s suggestion (also raised by **Rev 6ghW**) to examine the effects of violating this assumption. Accordingly, we have added experiments in **Appendix C.6** (of the revised version) showing the impact of both weak and strong violations of this assumption on our results.
>
> The main takeaway from these ablations is that both our method and previous methods relying on invariance (e.g., [3]) perform poorly when the assumption is violated. However, our method outperforms IRM [3] even in this adversarial setting and is competitive with other baselines when the assumption is only weakly satisfied (i.e., small heterogeneity across environments).
>
>
> **[W2 Selected subsets]** Due to the stochastic nature of optimization in our method, different subsets are selected for different initialization seeds.
>
>
> The most frequently selected features (>70% of the seeds)—maternal age, alcohol consumption, maternal foreign status, and number of prenatal care visits—align with the adjustment sets suggested in epidemiology literature.
>
> Less frequently selected features (<30% of the seeds) include marital status, firstborn status, and previous child mortality indicator (i.e. whether the mother had a child who died at birth). While a deeper epidemiological analysis would be needed, we suspect that the indicator for previous child mortality (which our method appropriately discards) is likely to be influenced by the treatment (i.e. smoking).
>
> We refrain from making strong claims about this real-world example, as this is beyond our expertise. *Our primary objective was to illustrate the practical use of our method*.
>
> Additionally, we emphasize that the main challenge in excluding bad controls based on domain knowledge is that, in this example, even experts cannot confidently determine whether certain factors were measured before or after smoking initiation.
>
> **[Q1 Descendant of $Y$]** Great question. A concise explanation is that $Y$ acts as a collider, and conditioning on a descendant of a collider introduces bias (as long as $T$  has a causal effect on $Y$). For a more formal and complete explanation of why descendants of $Y$ introduce bias, we refer to the discussion in [2] (see **Model 18**).
>
> **[Q2 Standard errors in application]**  Because our method and IRM rely on an optimization procedure to identify a valid adjustment set, they select different subsets depending on the random initialization, leading to high standard deviations across the 100 seeds. Table 1 shows that *IRM has a higher standard deviation than our method*. In contrast, the ALL and NULL baselines have low standard deviations, as they don’t rely on an optimization procedure to find the adjustment set (which is pre-specified).
>
> **[Q3 Generality of our approach]** Our method readily extends to other causal estimands, such as CATE and ATT. Extending our approach to continuous treatments may be more challenging, but we believe it is feasible.
>
>
> [1] Peters, Jonas, Peter Bühlmann, and Nicolai Meinshausen. "Causal inference by using invariant prediction: identification and confidence intervals." Journal of the Royal Statistical Society Series B: Statistical Methodology 78.5 (2016): 947-1012.
>
> [2] Cinelli, Carlos, Andrew Forney, and Judea Pearl. "A crash course in good and bad controls." Sociological Methods & Research 53.3 (2024): 1071-1104
>
> [3] Claudia Shi, Victor Veitch, and David Blei. Invariant representation learning for treatment effect estimation. Uncertainty in Artificial Intelligence, 2021.

---

> > ### Comment · Reviewer_r3F3 · 2024-11-22
> >
> > Thank you for your explanations. However, I would like to point out two remaining issues:
> > 1. The notations in the experiments are confusing and inconsistent, for example:
> > - In the problem setting and methodology sections, Z denotes the observed variables, d is the number of observed covariates, p is total number of variables. [d] is used to denote the indices of Z, but in Assumption 3.2, it refers to nodes.
> > - In the simulations, d is used inconsistently to denote the total number of nodes, the number of independent noises, and the association between the outcome and the descendant.
> > - Z is used to denote the descendant of T and Y in the appendix, corresponding to $X_c$ in Section 5.
> > - p is used as the subscript for the pre-treatment variable $X_p$ in Section 5, but it does not appear in the data generating process in the appendix.
> > - $\sigma$ is used to represent both a variance parameter and the sigmoid function in the data generating process.
> > - In the second row of Figure 9, the white space can be trimmed if the second RAMEN estimator is not used.
> > 2. Could you explain how $\sigma^2$ in Appendix C.6 introduces environment heterogeneity and how Assumption 4.1 is violated? It only shifts the means and amplifies the variance of observed variables within the same environment.

---

> ### Author Response · Authors · 2024-11-23
>
> Thank you very much for your feedback.
>
> 1. We greatly appreciate the time you took to point out the inconsistencies in the notation of the appendix. We have uploaded a revised version where the notation is adjusted accordingly.
>
>
> 2. Regarding the environment heterogeneity in Appendix C.6:
>    - For each environment, we sample $U \sim \mathcal{N}(0, \sigma^2 I_d)$ *only once*.
>
>    - Then, we sample $X_i \sim \mathcal{N}(U_i, 0.5 + U_i)$ for $i=1,\ldots, d_x$.
>    - If  $\sigma^2 = 0$, $U = 0$ becomes degenerate and $X \sim \mathcal{N}(0, 0.5)$ has *the same distribution across environments*. Therefore, there is no heterogeneity across environments and Assumption 4.1 is violated.
>    - If $\sigma^2 > 0$, the mean and variance of $X$ *will shift across environments*, with larger shifts as $\sigma^2$ increases. Therefore, the heterogeneity across environments increases as we increase the parameter $\sigma^2$, and Assumption 4.1 will be more likely to be satisfied.
>
> We hope this clarifies the issue and remain open to further questions or clarifications. We have described the data generating process more carefully in the revised version.

---

> > ### Comment · Reviewer_r3F3 · 2024-11-25
> >
> > Thank the authors for detailed responses, which have addressed most of my concerns.

---

### Author Response · Authors · 2024-11-17

We are grateful to the reviewers for their thoughtful and constructive comments that have improved the paper. We are pleased that they found our paper to be *well-written* (**r3F3, oUQ4**), to address an *important problem in causal inference*  (**r3F3**), and to include a *comprehensive experimental validation* (**BoRS, r3F3, oUQ4**).

While we have addressed the individual concerns of the reviewers in their respective threads, we summarize below what we think are the key issues raised and how we addressed them.

-----

## Strength of Assumption 4.1
The main concern raised by reviewers **r3F3** and **6ghW** was the strength of Assumption 4.1 and its applicability in practice.

In response, we conducted additional experiments (see **Appendix C.6**) to analyze the sensitivity of our method to violations of this assumption. The results show that our method outperforms the existing invariance-based methods, even in adversarial scenarios where Assumption 4.1 is fully violated. Moreover, in the more realistic scenario where Assumption 4.1 is weakly satisfied (i.e., small but non-zero heterogeneity across environments), our method outperforms all the other baselines.

Additionally, we remarked that Assumption 4.1 is *implied by well-established assumptions* in the causal invariance literature and *can be falsified* with a "small" amount of domain knowledge (e.g., if the practitioner knows a parent of $T$ or $Y$ in the covariate set).

---
## Lack of experiments with small sample size

Another concern raised by reviewer **BoRS** was the robustness of our method under varying sample sizes. To address this, we added new experiments (see **Appendix C.5**) to analyze the performance of our method when the sample size is small. These results show that *our method performs well even in small-sample regime*, outperforming the existing baselines.

---
## Inconsistencies in the notation

Reviewer **r3F3** pointed out some inconsistencies in the notation between the main text and the appendix.  In response, we have reviewed and adjusted the notation to ensure consistency throughout, and a revised version has been uploaded.

---

We believe these updates and our individual responses address the concerns raised by the reviewers and strengthen our paper. *We remain open to further feedback*.

---

### Meta-Review · Area_Chair_J5pG · 2024-12-20

**Metareview:**

The reviewers are in unanimous agreement to accept the paper with varying levels of enthusiasm.

Based on my own reading, I have the following additional comments:

-  "Post-treatment variable" normally may include observed mediators but the authors assume on such mediators can exist. Then the only post-treatment variables are colliders between treatment and the target? explicitly stating this early on may clarify confusion about applicability of the proposed method.

- The authors mention that adjustment usually requires causal graph. This is generally correct, but ignores the line of work which can do adjustment without the graph. For example,  Shah et al. "Finding valid adjustments under non-ignorability with minimal dag knowledge" is cited in Appendix under the expanded related work, but not in the main paper. I think in camera-ready version a more nuanced and precise discussion of the related work should be brought into the main paper from the Appendix to avoid any claims that might be misleading to the readers.

- As pointed out by the authors,  this paper assumes no post-treatment variables. A related work that allows post-treatment varibles is "Front-door Adjustment Beyond Markov Equivalence with Limited Graph Knowledge" by Shah et al. which the authors seem to have missed. Specifically, it would be interesting to compare the DAG knowledge assumed by these existing works with the causal knowledge assumed by the submitted paper.

- Assumption 3.3. simply says either the treatment or the target variable is not intervened across any environments. Might be good to voice this explicitly.

- On positivity assumption: "widely known to be necessary for identifying the treatment effect in observational studies". This is not technically correct. There is some recent work characterizing which positivity violations are OK, please see "On Positivity Condition for Causal Inference" by Hwang et al.

- "our double robustness property significantly differs from most classic results" I am not sure if a detailed account of the differences are provided in the manuscript. It is not very clear why the authors chose the name doubly-robust if it is so different from the existing results. A discussion needs to be added on this. Authors may also consider renaming the title to avoid confusing about double-robustness properties of their estimator.

**Additional Comments On Reviewer Discussion:**

Most reviewers seem satisfied with the author responses and rebuttal and have improved their scores.

---

### Decision · Program_Chairs · 2025-01-22

Accept (Poster)